# A Study of the Early-Stage Translations of *Foxing*佛性 in Chinese Buddhism: The *Da Banniepan Jing*大般涅槃經 Trans. Dharmakṣema and the *Da Fangdeng Rulaizang Jing*大方等如來藏經 Trans. Buddhabhadra

**Zijie Li** 



History School, Northwest University, Xi'an 710069, China; rblslzj@hotmail.com

**Abstract:** The *Da fangdeng rulaizang jing*大方等如來藏經 (Skt. *Tathāgatagarbha-sūtra*), translated by Buddhabhadra佛陀跋陀羅 (358–429) is one of the early Chinese Buddhist canon texts where the term *foxing*佛性 (Jp. *busshō*; Buddha-nature) is clearly used to express Buddha-nature. However, the term *foxing* cannot be confirmed in other extant translations of the *Tathāgatagarbha-sūtra*. Another early text in the Chinese Buddhist canon, the *Da banniepan jing*大般涅槃經 (Skt. *Mahāparinirvāṇa-mahāsūtra*), translated by Dharmakṣema曇無讖 (385?–433), also used the term *foxing*, which cannot be correspondingly confirmed in the surviving Sanskrit fragments of this scripture. Some significant differences in *foxing* between the Sanskrit fragments and Dharmakṣema's translation of this sutra belong to the first twelve fascicles of Dharmakṣema's translation completed under his collaborators' support when he had not mastered the Chinese language. It is very likely that Faxian法顯 (337–422) translated a version of the *Mahāparinirvāṇa-mahāsūtra* that featured *buddhadhātu* as *foxing*. Buddhabhadra, in the same period, translated a version of the *Tathāgatagarbha-sūtra*, in which he favoured the term *foxing* over a literal translation of the Sanskrit. As another contemporary monk with these two, Dharmakṣema translated the *Mahāparinirvāṇa-mahāsūtra*, going further than Faxian by using the term *foxing* regularly. These texts influenced the Dilun monastic tradition地論宗. Among these, the term *foxing* and its Sinicism explanations played the most significant role, influencing the whole of the Chinese and even East Asian Buddhist thought.

**Keywords:** *foxing*; *Da banniepan jing*; *Da fangdeng rulaizang jing*; Dharmakṣema; Buddhabhadra

## 1. Introduction

In recent years, discussions in Buddhist scholarship have focused on the concept of "Buddha-nature" within all sentient beings, whether or not this concept is compatible with classical Buddhist teachings such as no-abiding-self or even those doctrines rooted in the Nikāyas/Āgamas. This controversy is not only relevant to East Asian Buddhism but also to the roots of this tradition in the Indian Mahāyāna sutras, which deploy the concept of *tathāgatagarbha* (Buddha-embryo or Buddha-womb).[1] Moreover, in some cases, the term *tathāgatagarbha* is also used to describe sentient beings themselves (Skt. *sarvasattvās tathāgatagarbhāḥ*; all sentient beings are those who contain *tathāgata*). As is widely known, in the history of East Asian Buddhism, *tathāgatagarbha* (Chin. *rulaizang*如來藏) was sometimes considered a synonym of *foxing*佛性 (Jp. *busshō*; Buddha-nature).[2] The relationship between these two terms was ambiguous in Chinese Buddhism because some monks and schools, such as the Nirvāna tradition (Chin. *Niepan zong*涅槃宗), declared that *foxing* is the same as *rulaizang*.[3] Therefore, probing the early cases where the classical Chinese term *foxing* appeared in China is significant to clarifying the origin and development of these two concepts in East Asian Buddhism.

Two of the early translators who translated some terms as *foxing* were Buddhabhadra佛陀跋陀羅 (358–429) and Dharmakṣema曇無讖 (385?–433). Both of them worked on their

texts in China in the first half of the fifth century.[4] In this article, I investigate this issue based on the translations by Dharmakṣema and Buddhabhadra. In other words, cases of the term *foxing* that appeared during the Northern Liang北涼 dynasty (397–439) and the second half of the Eastern Jin東晉 (317–420) are the objects of this research.[5] Among these two dynasties, the Northern Liang is much more important for my discussion because the full text of the *Mahāparinirvāṇa-mahāsūtra* [the Great Nirvāṇa Sūtra] was completely translated into classical Chinese and spread to the whole of China after this period.[6]

There is hardly research discussing both the translations of Dharmakṣema and Buddhabhadra to probe the origin of the term *foxing* as a Chinese term and its context in Chinese translation in the early fifth century, especially the lack of comparison with relevant Sanskrit fragments in the context of Chinese Buddhism remains, although it is evident that in Indian Buddhist texts, *buddhadhātu* indicates Buddha-nature, which has the meaning "nature of a buddha."[7] The scholars in Indian and Tibetan Buddhist Studies did not pay attention to this issue in the context of East Asian Buddhism. Conversely, many scholars in Chinese Buddhist Studies have hardly used the relevant Sanskrit and Tibetan texts to investigate the origin and development of the term *foxing*.[8] In a sense, this is also one of the purposes of this article.

Through this study, we can presume that Faxian法顯 (337–422) translated a version of the *Mahāparinirvāṇa-mahāsūtra* that featured *buddhadhātu* as *foxing*. Buddhabhadra, in the same period, translated a version of the *Tathāgatagarbha-sūtra*, in which he favoured the term *foxing* over a literal translation of the Sanskrit, maybe a particular Sanskrit expression, or supplying *foxing* in place of diverse Sanskrit phrasings. As another contemporary monk with these two, Dharmakṣema translated the *Mahāparinirvāṇa-mahāsūtra*, going further than Faxian by using the term *foxing* regularly.

## 2. Dharmakṣema, Buddhabhadra and Chinese Buddhism at the Beginning of the Fifth Century

Unfortunately, most of the Buddhist literature produced during the Northern Liang dynasty has been lost. Only some quotes can thus be found from later treatises such as the *Da banniepan jing shu*大般涅槃經疏 [Commentary on the *Mahāparinirvāṇa-mahāsūtra*], written by Guanding灌頂 (561–632). In 421, it was recorded that Dharmakṣema held a Sanskrit text in hand and spoke Chinese to Daolang道朗 (?–?) when he translated the *Mahāparinirvāṇa-mahāsūtra* in 421. Daolang was one of the most accomplished monks in Hexi河西 area at that time and was guided by Dharmakṣema in Guzang姑臧 while receiving tuition from Dharmakṣema. According to the *Chu sanzang ji ji*出三藏記集 [Compilation of Notes on the Translation of the Tripitaka; or Collected Records concerning the Translation of the Tripiṭaka], 11 texts were regarded as Dharmakṣema's translations, as follows.[9]

*Da banniepan jing*大般涅槃經 [the Great Nirvāṇa Sūtra], 36 *juan*, T374.

*Fangdeng daji jing*方等大集經 [the Sūtra of the Vaipulya Great Assembly], 29 *juan* or 30 *juan*, T397.

*Fangdengwang xukongzang jing*方等王虛空藏經 [the Sūtra of the King of Vaipulya and the Chamber of Space], 5 *juan*.

*Fangdeng dayun jing*方等大雲經 [Skt. *Mahāmeghasūtra*; the Sūtra of the Vaipulya Great Cloud], 4 *juan* (or *Fangdeng wuxiang dayun jing*方等無想大雲經, 6 *juan*), T387.

*Beihua jing*悲華經 [the Sūtra of Flower with Compassion], 10 *juan*, T157.

*Jinguangming jing*金光明經 [the Golden Light Sūtra], 4 *juan*, T663.

*Hailongwang jing*海龍王經 [the Sūtra of the King of Marine Dragons], 4 *juan*, T598.

*Pusa dichi jing*菩薩地持經 [the Sūtra of Stages of Bodhisattvas], 8 *juan*, T1581.

*Pusajie ben*菩薩戒本 [the Text on Precepts of Bodhisattvas], 1 *juan* (also regarded as a text translated in Dunhuang燉煌), T1500.

*Youposai jie*優婆塞戒 [Upasaka's Precepts], 7 *juan*, *T*1488.

*Pusajie jing*菩薩戒經 [the Sūtra of Precepts of Bodhisattvas], 8 *juan*.

*Pusajie youpo jietan wen*菩薩戒優婆戒壇文 [the Treatise on Precepts of Bodhisattvas and Upasakas], 1 *juan*.

These eleven texts totally have 104 *juan*. In the dynasty of An emperor of Jin晋安帝, Indian monk Tanmochen曇摩讖[10] (or Tanwuchen曇無讖) came to Western Liang prefecture西涼州 and translated these texts under the support of Juqumengxun沮渠蒙遜.

Dharmakṣema moved to Guzang姑藏 and translated these texts after 417. In the *Chu sanzang ji ji*, the *Da banniepan jing*大般涅槃經 [the Great Nirvāṇa Sūtra] and the *Fangdeng daji jing*方等大集經 [the Sūtra of the Vaipulya Great Assembly] are mentioned. Among these, the most influential text in China is the *Da banniepan jing*.[11]

According to Fuse Kōgaku布施浩岳, if the first ten fascicles of the *Da banniepan jing* are compared with the *Nihuan jing*泥洹經 [the Nirvāṇa Sūtra], which is a short Chinese translation, or originally a short version, of the *Mahāparinirvāṇa-mahāsūtra* translated by Faxian法顯 (337–422) several years before the *Da banniepan jing* and has six fascicles (*Foshuo dabannihuan jing* 佛説大般泥洹經, T 376), it is evident that the *Da banniepan jing* contains most of the content and information that is found in the *Nihuan jing*, although it is possible that they are based on different versions/recensions of the *Mahāparinirvāṇa-mahāsūtra*. Accordingly, following the completion of the *Da banniepan jing*'s translation, fewer and fewer Chinese readers read the *Nihuan jing*.[12] This is a foundation of understanding the relationship between the *Da banniepan jing* and the *Nihuan jing* in Chinese Buddhism. Moreover, as it is well known, the southern text of the *Da banniepan jing*, containing thirty-six fascicles, was not a direct translation but an edited version based on the *Nihuan jing* translated by Faxian and the *Da banniepan jing* translated by Dharmakṣema. Faxian returned to China from India and launched his translation work after 410.[13] It almost overlapped with the date of the translation of the *Da banniepan jing*. Faxian cooperated with Buddhabhadra and finished the translation of the *Nihuan jing* in 416.[14] I will, therefore, ignore the southern text of the *Da banniepan jing* in this article. However, regarding the *Nihuan jing*, there will be a discussion of the *Nihuan jing*'s attitude to *foxing*, the purpose of my research, later in this article, since it is important for the study of what Dharmakṣema likely read.[15]

The original content of the *Nihuan jing* equates to the first ten fascicles of the *Da banniepan jing* translated by Dharmakṣema, which contains forty fascicles in total. After completing the first twelve fascicles of the translation of the *Da banniepan jing*, Dharmakṣema noted that the original Sanskrit text of the *Da banniepan jing* was not enough in China then; thus, he returned to India to seek an integrated version of this sutra. He arrived at Khotan于闐 by traveling through the southern path of Tianshan天山 and found the middle and later fascicles of the *Mahāparinirvāṇa-mahāsūtra*.[16]

Concerning the Sanskrit text, Takasaki Jikidō高崎直道 mentioned partial fragments of this text.[17] Almost simultaneously, Matsuda Kazunobu松田和信 collected and translated all existing fragments[18], and Habata Hiromi幅田裕美 used the previous research by Takasaki and Matsuda and provided a critical edition of the Tibetan text[19], followed by a published monograph on all extant Sanskrit fragments of the *Mahāparinirvāṇa-mahāsūtra*.[20]

Ōno Hōdō大野法道 thinks the preface (Chin. *jingxu* 經序) by Hexi Daolang河西道朗 is more reliable. According to this account, Dharmakṣema first came to Dunhuang 燉煌 with a variety of scriptures, and then moved to the Northern Liang 北涼, where he translated the first ten *juan* of the *Da banniepan jing*, finishing in 421. He further states that it is certain that the source text for the remaining thirty *juan* incorporated into Dharmakṣema's the *Da banniepan jing* came from Khotan.[21] According to Ōchō Enichi橫超慧日, conceiving that the same person or group translated at the same time all forty fascicles of the *Da banniepan jing* is difficult. Instead, they may have been edited by different people over several stages.[22]

In other words, some fascicles or sections of the forty-fascicle *Da banniepan jing* were not translated by Dharmakṣema. Feng Chengjun馮承鈞 states that Dharmakṣema used the Sanskrit text collected by Zhimeng智猛 (?–452), who did not engage in translating.[23] In contrast, Chen Jinhua陳金華 objects to Feng's conclusion. Chen asserts that Zhimeng not only brought the Sanskrit text of the *Da banniepan jing* to China but also participated in the translation.[24] Hence, it is very likely that the translators of the *Da banniepan jing* were not only Dharmakṣema but included other people or groups, in addition to even some other materials or sources.[25]

Therefore, I intend to discuss the classical Chinese term *foxing* based on the *Da banniepan jing* attributed to Dharmakṣema in the framework of Chinese Buddhism. In addition, as a background to contemporary translation in China, the term *foxing* as found in the *Da fangdeng rulaizang jing*大方等如來藏經 [The Sutra of the Tathāgatagarbha] translated by Buddhabhadra佛陀跋陀羅 (358–429) is also the object of my discussion. In other words, as these two texts were translated into classical Chinese during almost the same and early period, I will talk of the use of the term *foxing* found in the *Da fangdeng rulaizang jing* at first and then move to the discussion on this topic in the *Da banniepan jing*.[26] Finally, this study also slightly mentions the *Da fangdeng wuxiang jing* 大方等無想經 [Skt. *Mahāmeghasūtra*; T387] by Dharmakṣema and the renderings by Guṇabhadra 求那跋陀羅 (394–468) from roughly the same time.

According to the *Fozu tongji*佛祖統紀 [Entire Records of the Buddhas and the Ancestors], both Dharmakṣema and Buddhabhadra were engaged in the translation works in China during almost the same era.

In the fourth year of Yixi義熙, Huiyuan慧遠 was discontented with the uncompleted translations in Chinese Buddhism, the lack of meditation, and the incomplete canons of precept. He sent his disciples such as Zhifaling支法領 to India to collect more Sanskrit Buddhist texts. They met Buddhabhadra in India and asked him to return to China together. In the eighth year of Yixi, Dharmakṣema moved to Guzang姑臧. Juqu Mengxun沮渠蒙遜, the King of the Northern Liang, asked Dharmakṣema to stay there and translate the *Mahāparinirvāṇa-mahāsūtra* into the *Da banniepan jing*, which has forty fascicles. In the ninth year of Yixi, Buddhabhadra (Chin. Juexian覺賢), a monk from the Kapilavastu area, went to Lushan廬山 mountain and stayed there. Huiyuan asked Buddhabhadra to translate some texts of meditation.

義熙四年，遠法師以江東經卷未備，禪法無聞，律藏殘缺。乃令弟子支法領等往天竺，尋訪獲梵本。於于闐遇佛陀跋陀羅，乃要與東還。… 八年，曇無讖至姑臧，涼王沮渠蒙遜留之譯《大般涅槃經》四十卷。… 九年，迦維衛國沙門佛陀跋陀羅（此雲覺賢）至廬山入社，遠法師請譯禪數諸經。[27]

That is to say, Huiyuan's 慧遠 (334–416) disciples met Buddhabhadra in about 408. In 412, Dharmakṣema launched the translation of the *Da banniepan jing* in the Northern Liang. In 413, Buddhabhadra had met Kumārajīva鳩摩羅什 (344–413) in China and moved to Lushan Mountain to continue his translation works. Evidently, the periods of Dharmakṣema and Buddhabhadra acting in China overlapped. Therefore, in my opinion, it is very likely that they had similar circumstances from 410, such as mutual assistants in China and the same Chinese texts that had been translated.[28] On the contrary, to say the least of it, even if these two translators did not actually share many assistants and they did not directly influence each other, we cannot deny that they were engaged in the translation works in China during the same era, a very early period for the appearance of the term *foxing*. For this reason, it is inevitable to discuss not only Dharmakṣema's translations but also those by some other translators, such as Buddhabhadra, at the beginning of the fifth century.[29]

As has been mentioned above, there is hardly any research discussing both the translations of Dharmakṣema and Buddhabhadra to probe the origin of the term *foxing* as a Chinese term and its context in Chinese translation in the early fifth century, especially the

lack of comparison with relevant Sanskrit fragments in the context of Chinese Buddhism remains. This is also one of the purposes of this article.

### 3. *Foxing*佛性 in the *Da fangdeng rulaizang jing*大方等如來藏經 (Skt. *Tathāgatagarbha-sūtra*) Translated by Buddhabhadra

One of the earliest Buddhist texts discussing *tathāgatagarbha* is the *Tathāgatagarbha-sūtra*. The *Tathāgatagarbha-sūtra*, a seminal text of *tathagatagarbha* doctrine, describes how *tathāgatagarbha* accounts for the possibility of transformation from a state of delusion to a state of enlightenment by uncovering the inherently pure nature within, referred to containing, store or "that which sentient beings possess."[30]

Two recensions of the *Tathāgatagarbha-sutra* are extant in Chinese: the *Da fangdeng rulaizang jing*大方等如來藏經 (*T* vol. 16, no. 666), translated by Buddhabhadra佛陀跋陀羅 (358–429) in the Eastern Jin東晉 (317–420),[31] and the *Da fangguang rulaizang jing*大方廣如來藏經 (*T* vol. 16, no. 667), translated by Amoghavajra (or Bukong) 不空 (705–774) under the Tang唐 (618–907).[32] While the original Sanskrit sutra is not extant for comparison, the bKa' 'gyur canon represents Tibetan recensions;[33] one of them is titled *Phags pa de bzhin gshegs pa'i snying po shes bya ba thegs pa chen po'i mdo*, translated by Śākyaprabha and Ye-śes-sde (photographic print Tibetan Buddhist Canon 36, 240.1~245.5)[34].

The *Tathāgatagarbha-sūtra* is a relatively short scripture that represents the point of a number of works in Indian Buddhism concentrating on the idea that all sentient beings are *tathāgatagarbha*. According to Michael Zimmermann, the hitherto accepted assumption that the *Da fangdeng rulaizang jing* reflects an Indian transmission, which has not undergone the textual alterations of later centuries, is only partly true because the source of the citations in the *Ratnagotravibhāga* [Chin. *Jiujing yisheng baoxing lun*究竟一乘寶性論; Treatise of the Jewel-nature of Ultimate Single Vehicle], a *śāstra* which was written at least fifty years before Buddhabhadra translated *Tathāgatagarbha-sūtra* into the *Da fangdeng rulaizang jing*, has turned out to be the recension represented in the Tibetan tradition.[35] I agree with this view. That is to say, although it is possible that there are some differences between the underlying Sanskrit text of the *Da fangdeng rulaizang jing* and that of the Tibetan translation, we should not consider that the original Sanskrit text of the extant Tibetan translation had been substantially amended compared with that of the *Da fangdeng rulaizang jing*. Thus, in my opinion, it is still effective to investigate the unique way and purpose of Buddhabhadra through comparing his translation with the Tibetan text, especially about the Chinese term *foxing*, which appears only in Buddhabhadra's translation.

As mentioned above, the *Da fangdeng rulaizang jing* translated by Buddhabhadra is a text early in the known literary history of the term *foxing*. At the beginning of this chapter, the following paragraph must be discussed prior to others.

In the same way, sons of good family, I see with my buddha-vision that all sentient beings, within the afflictions of desire, hostility and delusion, possess the knowledge, vision and body of a *tathāgata* sitting cross-legged, dignified and motionless. Sons of good family, all sentient beings, although situated in many kinds of rebirth, in the midst of their afflictions possess the *tathāgatagarbha*, always permanent and undefiled, replete with excellent characteristics no different to my own. Sons of good family, for example, it is like this, a person with divine-vision/eye (Skt. *divyacakṣus*; Chin. *tianyan* 天眼) inspects calyxes and find that the *tathāgata*-body within the flowers can be revealed if the drooped petals had been moved away. In the same way, sons of good family, the Buddha, seeing all sentient beings to already be *tathāgatagarbha*, desiring to cause this to be revealed, explains the dharma, destroying their defilements and manifesting their buddha-nature. Sons of good family, the true nature (Skt. *dharmatā*; Chin. *faer*法爾) of all buddhas is this: whether or not buddhas appear in the world, in all living beings the store [or womb] of a *tathāgata* is at all times present without change.

> 如是善男子！我以佛眼觀一切眾生，貪欲恚癡諸煩惱中，有如來智、如來眼、如來身，結加趺坐儼然不動。善男子！一切眾生雖在諸趣，煩惱身中有如來藏，常

無染污、德相備足，如我無異。又善男子！譬如天眼之人觀未敷花，見諸花內有
如來身結加趺坐，除去萎花便得顯現。如是善男子！佛見眾生如來藏已，欲令開
敷為説經法，除滅煩惱顯現佛性。善男子！諸佛法爾，若佛出世若不出世，一切
眾生如來之藏常住不變。[36]

[Tibetan translation] In the same way, sons of good family, also the Tathagata, the Honorable One and Perfectly Awakened One, [perceives] with his insight *(prajñā)*, knowledge *(jñāna)* and tathāgata-vision that all the various sentient beings are encased in myriads of defilements, [such as] desire *(rāga)*, anger *(dveṣa)*, misguidedness *(moha)*, longing *(tṛṣṇā)* and ignorance *(avidyā)*. And, sons of good family, [he] perceives that inside sentient beings encased in defilements sit many tathāgatas, cross-legged and motionless, endowed like myself with a [tathāgata's] knowledge and vision. And [the Tathāgata], having perceived inside those [sentient beings] defiled by all defilements the true nature of a tathāgata *(tathāgatadharmatā)* motionless and unaffected by any of the states of existence, then says: "Those tathāgatas are just like me!" Sons of good family, in this way a tathāgata's vision is admirable, [because] with it [he] perceives that all sentient beings contain a tathāgata *(tathāgatagarbha)*. "Sons of good family, it is like the example of a person endowed with divine vision [who] would [use this] divine vision to look at such unsightly and putrid lotuses, not blooming and not open, and would [owing to his vision] recognise that there are tathagatas sitting cross-legged in their center, in the calyx of [each] lotus, and [knowing that, he] would then desire to look at the forms of the tathāgatas; [he would] then peel away and remove the unsightly, putrid and disgusting lotus petals in order to thoroughly clean the forms of the tathāgatas. In the same way, sons of good family, with the vision of a buddha, the Tathāgata also perceives that all sentient beings contain a tathāgata *(tathāgatagarbha)*, and [therefore] teaches the Dharma [to them] in order to peel away the sheaths of those sentient beings [encased in such] defilements [as] desire, anger, misguidedness, longing and ignorance. And after [those sentient beings] have realized the [Dharma, their] tathāgatas [inside] are established in the perfection [of the tathāgatas] *(ma rig pa'i nyon mongs pa'i sbubs dbye ba'i phyir chos ston te/de sgrub pa'i de bzhin gshegs pa rnams ni yang dag pa nyid du gnas so)*." Sons of good family, the essential law *(dharmatā)* of the *dharmas* is this[37]: whether or not tathāgatas appear in the world, all these sentient beings at all times contain a tathāgata *(tathāgatagarbha)*.[38]

It is notable that in the Tibetan translation, there is a sentence stating "*de bzhin gshegs pa rnas ni yan dag pa nyid du gnas so*," rather than the *Da fangdeng rulaizang jing* stating "*xianxian foxing*顯現佛性 (manifesting Buddha-nature)." In other words, in the Tibetan translation, it is difficult to identify the term corresponding to *foxing* in this paragraph, while it is very possible that this Tibetan translation is not translating the same Indic text as Buddhabhadra.

A similar example can also be found in the following passage.

They [women] do not know it. Therefore, the *Tathāgata* teaches widely the Dharma for living beings saying: "Sons of good family, do not denigrate yourself! In your own body, you all have the buddha-nature. If you practice diligently and diminish all evil, then you will attain the designations 'bodhisattva' and 'exalted one.' You will guide and save innumerable living beings!"

如彼女人而不覺知，是故如來普為説法，言：善男子！莫自輕鄙，汝等自身皆有
佛性，若勤精進滅眾過惡，則受菩薩及世尊號，化導濟度無量眾生。[39]

[Tibetan translation] Then, though the element of a *tathāgata* has entered into sentient beings and is present within, those sentient beings do not realize [it]. Sons of good family, in order that sentient beings do not despise themselves, the Tathāgata in this [connection] teaches the Dharma with the [following] words: "Sons of good family, apply energy without giving in to despondency! It will happen that one day the tathāgata [who has] entered [and] is present within you will become manifest. *(rig kyi bud ag khyed bdag nyid sro shi bar ma byed par khyed brtson 'grus brtan par gyis shig dang/khyed la de bzhin gshegs pa zhugs pa yod pa dus shig na 'byung bar 'gyur te)* Then you will be designated "bodhisattva," rather than

"[ordinary] sentient being (*sattva*)." [And] again in the [next stage you] will be designated "buddha," rather than "bodhisattva"."[40]

Here, Buddhabhadra translated "*rudeng zishen jie you foxing*汝等自身皆有佛性 (you all have Buddha-nature)," compared to the statement, "it will happen that one day the Tathāgata who has entered and is present within you will become manifest. Then you will be designated a bodhisattva, rather than ordinary sentient being (*sattva*)" in the Tibetan translation. We can only find *tathāgata* and *sattva* in the Tibetan translation, rather than a proper term matching the Chinese term *foxing*.

Similarly, the following passage is also typical of the difference between these two translations.

In the same way, with the vision of a Sugata (buddha) I can see that although living beings are covered over by defilements, their *tathāgata*-nature is indestructible. I teach the Dharma with appropriate means in order to let living beings attain buddhahood. Because their buddha-nature has been covered by defilements, I intend to remove the defilements to make their buddha-nature purified rapidly.

善逝眼如是，觀諸众生類，

煩惱淤泥中，如來性不壞。

隨應而説法，令辦一切事，

佛性煩惱覆，速除令清淨。 [41]

[Tibetan translation] In the same way I can see that also all sentient beings have for a long time been constantly overpowered by defilements, but knowing that their defilements [are only] accidental (*āgantuka*), [I] teach the Dharma with [appropriate] means in order to purify [their] intrinsic nature (*prakṛti*). (*de dag gi nib lo bur nyon mongs shes/rang bzhin sbyang phyir thabs kyis chos ston to*)[42]

The term *foxing*佛性 appears in Buddhabhadra's translation again. If we check the Tibetan translation, the corresponding term for the Tibetan is likely to be "*prakṛti* (intrinsic nature)." There does not seem to be a term corresponding to at least the Chinese character *fo*佛.[43]

In particular, among Chinese renderings of the *Tathāgatagarbha-sūtra*, the classical Chinese term *foxing* can only be found in the *Da fangdeng rulaizang jing*, which was translated by Buddhabhadra. It is difficult to accurately confirm the relevant term for *foxing* or corresponding Tibetan terms, such as *sangs rgyas kyi khams/dbyings*, in both Amoghavajra's classical Chinese and the Tibetan translation. Similarly, as mentioned above, there are various terms related to the Chinese term *rulaizang* in the Tibetan translation, rather than a fixed term.

As is argued by Zimmermann (2002) and some other scholars, since it seems that Buddhabhadra has translated a different recension of the *Tathāgatagarbha-sūtra*, the fact that its content is different should not surprise us. I also concur with this opinion. The reason for discussing the *Da fangdeng rulaizang jing* by Buddhabhadra here is to reconsider the importance of this rendering for the history of the term *foxing* and Buddha-nature thought in Chinese Buddhism—that is, either the different Sanskrit recension of the *Tathāgatagarbha-sūtra*, which was read and used by Buddhabhadra, or Buddhabhadra's own creation influenced Chinese Buddhist thought at an early stage. This point was, to my knowledge, seldom emphasized by scholars in East Asian Buddhist studies.

To summarize, although a lack of clarity about Buddhabhadra's reasoning and motivation remains, as an early classical Chinese Buddhist canon text, the *Da fangdeng rulaizang jing* used the term *foxing*, which cannot be confirmed in other extant translations of the *Tathāgatagarbha-sūtra*.

#### 4. *Foxing*佛性 in the *Da banniepan jing*大般涅槃經 (Skt. *Mahāparinirvāṇa-mahāsūtra*) Translated by Dharmakṣema

The universality of emptiness (*śūnyatā*) and the doctrines of no-abiding-self (*anātman*) and impermanence (*anitya*) are some basic Buddhist teachings. Conversely, one still finds texts such as the *Śrīmālādevī-sūtra* and the *Mahāparinirvāṇa-mahāsūtra* that use terms such as *ātman*.[44] This is, in a sense, one of the most basic issues in Buddhist Studies.[45] This section focusses on which term the translators used to express the meaning such as *ātman* in the Chinese translation of the *Mahāparinirvāṇa-mahāsūtra*.

Before discussing Dharmakṣema's translation, Faxian's rendering should be mentioned first. Faxian's version seems to be a short Chinese translation of the *Mahāparinirvāṇa-mahāsūtra* if we merely read the Chinese translations, but it is better understood to be a Chinese translation of a shorter version of the *Mahāparinirvāṇa-mahāsūtra*. Following Habata, the term *sangs rgyas kyi khams*, which is a translation of *buddhadhātu* in the Tibetan rendering, is found 23 times in the Tibetan version of the *Mahāparinirvāṇa-mahāsūtra*. This number is relatively small because we find many instances of *foxing* in the Chinese translations. Among these 23 references, Dharmakṣema translates 17 instances of *foxing*, compared to only 8 of *foxing* in Faxian's rendering. Moreover, there is no example where Faxian translates *sangs rgyas kyi khams* as *foxing* but Dharmakṣema does not.[46] For this reason, I will focus on Dharmakṣema's rendering in this section.

Takasaki Jikidō notes that the underlying term of the *foxing* in the *Da banniepan jing*, translated by Dharmakṣema, refers to the nature of *tathāgata* (Chin. *rulai*如來).[47] Both Shimoda Masahiro下田正弘 and Michael Radich state that in the *Mahāparinirvāṇa-mahāsūtra*, a strong connection exists between *buddhadhātu* (Buddha-nature) and *tathāgatagarbha* (the embryo of Buddha), related to *stūpa* (relic-chamber).[48] Kanō Kazuo加納和雄 also asserts that both *buddhadhātu* and *tathāgatagarbha* refer to the content of a *stūpa*. Furthermore, two kinds of meaning in *dhātu*, containing both body and relics, are present. Beings possess *buddhadhātu*, understood as a Buddha's relic, which evokes the interior of a *stūpa* at which a relic generally sat.[49] Saliently, the term *buddhadhātu* (Chin. *foxing*), regarded as the most significant term in the *Da banniepan jing* (Skt. *Mahāparinirvāṇa-mahāsūtra*), cannot be found in the extant Sanskrit fragments of this scripture.[50] *Buddhadhātu*, as noted by Takasaki and Radich, in the *Mahāparinirvāṇa-mahāsūtra*, is considered a synonym of *tathāgatagarbha*. Alternatively, strictly speaking, *tathāgatagarbha* may be a way of referring to the presence of *buddhadhātu*. Meanwhile, we should not ignore the cases where the Chinese term *foxing* is not explained by referring to *tathāgatagarbha*.[51]

An interesting fact appears: the classical Chinese term *foxing*, emphasized in various classical Chinese translations of the *Mahāparinirvāṇa-mahāsūtra*, cannot be found in the existing Sanskrit fragments of the *Mahāparinirvāṇa-mahāsūtra*.[52] Therefore, the statement that in the *Mahāparinirvāṇa-mahāsūtra*, a strong connection existing between Buddha-nature and *tathāgatagarbha*, which was pointed out by Shimoda and Radich, is mainly based on the Tibetan and Chinese translations of the *Mahāparinirvāṇa-mahāsūtra*. In this case, it is meaningful to reconsider the original terms and the reasons they were translated into the Chinese term *foxing* by these translators, including Dharmakṣema.

Kamalaśīla (ca. 740–797) quotes the following in the *Bhāvanākrama*.

Thus, the *Mahāparinirvāṇa-mahāsūtra* states the following. Śrāvakas fail to see the lineage (*rigs*; gotra) of *tathāgata* in themselves because their meditation (*samādhi*) is strong, compared to their weak wisdom. Bodhisattvas can merely see an undefined lineage of *tathāgata* because their wisdom is strong, compared to their weak *samādhi*. *Tathāgata* can see all of these because he possesses both meditation and wisdom.

> de'i phyir 'phags pa yongs su mya ngan las 'das pa chen po'i mdo las kyang nyan thos rnams kyis ni de bzhin gshegs pa'i rigs mi mthong ste/ting nge 'dzin gyi shas che ba'i phyir dang/shes rab chung ba'i phyir ro//byang chub sems dpa' rnams kyis ni mthong mod kyi mi gsal te/shes rab kyi shas che ba'i phyir dang/ting nge 'dzin chung ba'i phyir ro//de bzhin gshegs pas ni thams cad

gzigs te/zhi gnas dang lhag mthong mtshungs par ldan pa'i phyir ro zhes bska' stsal te/[53]

The bodhisattvas of the ten abodes possess strong wisdom but little *samādhi* (meditation), so that they cannot clearly see *foxing* (Buddha-nature). Śrāvakas and pratyekabuddhas possess strong *samādhi* but little wisdom, so that they cannot clearly see Buddha-nature. Buddhas can clearly see Buddha-nature because they have both meditation and wisdom and have achieved buddhahood without any obstacle.

十住菩薩智慧力多，三昧力少。是故不得明見佛性。声聞縁覚三昧力多，智慧力少。以是因緣不見佛性。諸佛世尊定慧等故，明見佛性，了了無礙。[54]

Commenting on this, Yoshimura Shūki芳村修基 contends that Kamalaśīla's quote corresponds to the thirty-first fascicle of the *Da banniepan jing* translated by Dharmakṣema.[55] Except for small differences in their depictions of the śrāvaka and bodhisattva, these two translations correspond with each other very well. Matsuda denied the possibility that Kamalaśīla knew about the existence of the *Da banniepan jing* translated by Dharmakṣema.[56] If so, Kamalaśīla merely employed the Sanskrit text and Tibetan translations to quote the sentence that śrāvakas fail to see the lineage (*rigs*; gotra) of *tathāgata*. Conversely, the *Da banniepan jing*, translated by Dharmakṣema, clearly states that śrāvakas and bodhisattvas cannot see *foxing* (Buddha-nature). Hence, the lineage (*gotra*) found in this Tibetan quote was translated as *foxing* in the *Da banniepan jing*.[57]

Some similar cases appear in the *Jiujing yisheng baoxing lun*究竟一乘寶性論 [Skt. *Ratnagotravibhāga*]; namely, *gotra* was translated as *zhenru foxing*真如佛性 (Buddha-nature in thusness).

In summary, all beings, according to the Buddha, are always *tathāgatagarbha* according to three meanings: the *tathāgata*'s *dharmakāya* (Dharma-body) is omnipresent in all beings; there is no difference in the *tathāgata*'s *tathatā* (thusness); and the *gotra* of *tathāgata* (the cause for Buddhahood) exists.

samāsatas trividhenārthena sadā sarvasattvās tathāgatagarbhā ity uktaṃ bhagavatā/yad uta sarvasattveṣu tathāgatadharmakāyaparispharaṇārthena tathāgatatathatāvyatibhedārthena tathāgatagotrasaṃbhavārthena ca/(RG, 26, 7–9)[58]

This passage indicates three meanings. *Tathāgata*, therefore, the Tathāgata taught that all beings always have and share the embryo of Buddha (Skt. *tathāgatagarbha*; Chin. *rulaizang* 如來藏). What are these three kinds? First, *tathāgata*'s *dharmakāya* (Dharma-body) is omnipresent in all beings. It is said *fo fashen bianman*佛法身遍滿 (Dharma-body of Buddha is omnipresent). Second, there is no difference in *tathāgata*'s *zhenru* (thusness). It is said *zhenru wu chabie*真如無差別 (there is no difference in thusness). Third, all beings have *zhenru foxing*真如佛性. It is said *jie shiyou foxing*皆實有佛性 (all beings possess Buddha-nature).

此偈明何義。有三種義，是故如來說一切時一切眾生有如來藏。何等為三？一者，如來法身遍在一切諸眾生身，偈言佛法身遍滿故。二者，如來真如無差別，偈言真如無差別故。三者，一切眾生皆悉實有真如佛性，偈言皆實有佛性故。[59]

In the Sanskrit text, the third part of the definition of *tathāgatagarbha* is *tathāgatagotra*. In the classical Chinese translation, this term is translated as *zhenru foxing*真如佛性. Crucially, the three parts of *tathāgatagarbha*'s definition, namely, *dharmakāya*, *tathatā* and *gotra*, have been modified in the classical Chinese translation into: *fashen*法身・*zhenru*真如・*zhenru foxing*真如佛性, respectively.[60] This kind of translation of *gotra* in the *Jiujing yisheng baoxing lun*[61], translated by Ratnamati勒那摩提 (6th century CE; ?-508-?), is the same as in the *Da banniepan jing* translated by Dharmakṣema.[62] The *Jiujing yisheng baoxing lun* was translated into classical Chinese nearly a century after the *Da banniepan jing*. Furthermore, both were translated during the northern Chinese dynasties. For these reasons, the monks of Dilun地論 tradition were very likely to have been influenced by the terms and concepts of the *Da banniepan jing*.

On the other hand, due to the edition and research on the Sanskrit texts of the *Ratnagotravibhāga* and the *Laṅkāvatāra-sūtra*, it has been clarified that *yichanti*一闡提 (Skt. *icchantika*; beings who cannot achieve the buddhahood) is the translation of the Sanskrit term *icchantika*. Saliently, as Mizutani noted, the term *icchantika*, which was used in the *Mahāparinirvāṇa-mahāsūtra* at a very early stage, cannot be found in any surviving Buddhist scripture established prior to the *Mahāparinirvāṇa-mahāsūtra*.[63] In East Asia, numerous monks and scholars have attempted to demonstrate the possibility that *icchantika* achieve buddhahood.[64] The most important issue, however, is the controversy about *gotra* and Buddha-nature (Chin. *foxing*).

As introduced above, Matsuda edited the existent Sanskrit fragments of the *Mahāparinirvāṇa-mahāsūtra*, which can be used to further research the classical Chinese translation. In addition, Habata Hiromi edited the extant Sanskrit fragments and provided a new translation in 2019, which is more in-depth. In these Sanskrit fragments, there is one section, as stated below.

*Icchantikas* do not see (*na paśyanti*) virtuous deeds (*kalyāṇakṛta*). They see blame and evil (*papa*). Virtuous deeds (*sukṛta*) mean Bodhi (or enlightenment).[65] Not coming means not approaching. The esoteric (or intended) meaning means what is virtuous (*kalyāṇa*). Who is far away from esoteric deeds (*sandhākarma*)? Auspicious deeds (*bhadrakarma*) do not approach the *icchantika*. Who is far away from a good mind? A good mind does not approach *icchantika* because they are not wholesome beings due to their arrogant attitudes. What is the basic branch (*mūlāṃga*)? It means abandoning (or rejecting) this sutra (*sūtrapratikṣepa*). It is terrible because abandoning (or rejecting) the sutra is frightful. . . . . . . Who does not see (or understand) virtuous deed (*kṛta*)? Evil *icchantika* does not understand virtuous deed. *Icchantika*s do not see (or understand) virtuous deed until the end of their transmigration. I will summarize these meanings. Therefore, we should take these terrible things seriously because it is of the utmost frightfulness. At the time when all beings, after having become of one mind, will recognize the ultimate enlightenment (*anuttarāṃ saṃmyaksaṃbodhi*); it will be possible for *icchantika* to recognize Bodhi (enlightenment) at that time. However, *icchantika*s do not see (or understand) virtuous deed. The people who does not see enlightenment and virtuous deed should understand the fact. Namely, the deed of the Tathāgata will not end (or destroyed) unless all beings involved in transmigration recognize the ultimate enlightenment. At that moment, the Buddha will come to complete final *nirvāṇa*. Along with the final *nirvāṇa* (*atyantaparinirvāṇa*), the Buddha will become changeable and absent, like fire and a lamp.[66]

icchaṃti[kāḥ ka]lyāṇakṛtaṃ na paśyata(?): paśya(ṃ)ti tu pāpaṃ nidiśtuṃ(!) garhituṃ(!) [ca] . . .

sukṛ(ta)[ṃ b]o[dh]i[r] it[y] arthaḥ (/) na vyaiti n[ā]gacchatīty arthaḥ (/) [san]dheti kalyāṇam ity arthaḥ (/) saṃndhākarmaviśiṣṭakalyāṇaṃ kasya nāgacchati (/) bhadrakarma icchaṃtikasya nāga[ccha]ti (/) +++++++ laṃ satva icchaṃtikā iti i. +++++++++++ [ki]ṃ mūlā(ṃ)gaṃ sūtrapratikṣepaḥ (/) [ta]smād bhetavyaṃ sūtrapratikṣepako hi dāruṇaṃ . . .

[kaḥ] kṛtaṃ na paśyati (/) saṃsārakoṭyāṃ sa na paśyati (/) arthaṃ bhāṣiṣye: [saṃkṣepa]samuccayaṃ tasmād bhetavy[āḥ] (pa)[ra]madāru○ṇā[ḥ] (/) yadā sa[rvbasa]tvā [e]kamanaso bhūtvā anuttarāṃ saṃmyaksaṃbodhim abhisaṃbotsyate.

tadā [i]cchaṃ[tika](ḥ pā)po 'pi saṃ(bo)[tsyate] . . .dā [pa○rāṃ] bodhi(ṃ /) sarvba[kṛ](taṃ sa) na paśyati evaṃ jānīṣva viśārada (/) kasya kṛtaṃ na paśyati tathāgatas[y]a (/) [yadā] sa[rvbasa]tvā anuttarāṃ saṃmyaksaṃbodhim abhi○saṃbotsyate saṃsāra . . . [tā] tadā tathāgatasya kṛtaṃ na vinakṣya(ti/ta)dā [par]inirvbāyātyaṃtaparinirvbāṇe[na anit]yo bu[ddho bhaviṣya]ti [d]īpa ivendhana . . .ā[dagdhir iva . . . /[67]

Good man, [regarding *icchantika*s,] "not seeing" refers to not seeing the buddha-nature. "What is good" is *anuttarā samyaksaṃbodhi* itself. To say "they will not do it" refers to

[an *icchantika*] being unable to approach a good friend who can guide him spiritually. "Only seeing" refers to seeing without good reason. The word "bad" here refers to their repudiation of the well-balanced Mahāyāna sutras. And "this they may do" refers to the fact that *icchantika*s do say there are no well-balanced [sutras]. The meaning of the verse is simply that *icchantika*s do not think in a way that advances them toward the pure and good dharma. What is the pure and good dharma? It is nirvāṇa itself! To advance toward nirvāṇa refers to the capacity to cultivate practices that are wise and good, yet *icchantika*s have no practices that are wise and good. This is why they are incapable of progressing toward nirvāṇa. "On that basis, one should be afraid" refers to repudiating the true-dharma. Who should be frightened? ... In addition, one may also speak of "not seeing what has been done" in reference to the fact that *icchantika*s do not admit to themselves the host of bad things they have done. Because the *icchantika*s are arrogant, even though they often do things that are harmful, while doing them they initially have no sense of fear. This is why *icchantika*s are unable to attain nirvāṇa; they are like monkeys grabbing at the [reflection of the] moon in the water. Good man, if all living beings, however innumerable, were to all at once attain *anuttarā samyaksaṃbodhi*, the tathāgatas would still not see the *icchantika*s attaining *bodhi*. This is also the meaning of what I have called "not seeing what has been carried out." Furthermore, not seeing whose deeds were carried out means not seeing that carried out by the Tathagata. The Buddha has expounded the existence of buddha-nature for the benefit of living beings, but *icchantika*s transmigrate through *saṃsāra* unable to discern what this is. It is in this sense that I used the phrase "not noticing what has been done by the tathāgatas." *Icchantika*s will also look at the complete nirvāṇa of the Tathāgata and say to themselves, "This truly shows impermanence, nothing more than a lamp going out when its oil is extinguished."[68]

不見者謂不見佛性。善者即是阿耨多羅三藐三菩提。不作者所謂不能親近善友。唯見者見無因果。惡者謂謗方等大乘經典。可作者謂一闡提説無方等。以是義故，一闡提輩無心趣向清淨善法。何等善法。謂涅槃也。趣涅槃者謂能修習賢善之行。而一闡提無賢善行，是故不能趣向涅槃。是處可畏謂謗正法，誰應怖畏？...... 復次不見所作者謂一闡提所作衆惡而不自見。是一闡提憍慢心故，雖多作惡，於是事中初無怖畏。以是義故，不得涅槃。喻如獼猴捉水中月。善男子，假使一切無量衆生一時成於阿耨多羅三藐三菩提已，此諸如來亦復不見彼一闡提成於菩提。以是義故，名不見所作。又復不見誰之所作，所謂不見如來所作。佛為衆説有佛性，一闡提輩流轉生死，不能知見。以是義故，名為不見如來所作。又一闡提見於如來畢竟涅槃，謂真無常。猶如燈滅，膏油俱尽。[69]

The Sanskrit fragment states that *icchantika*s do not see (or understand) virtuous deeds (*kalyāṇakṛta*). Since this Sanskrit fragment is very likely later than what Dharmakṣema would have translated, we cannot know that this Sanskrit was what Dharmakṣema translated. The only thing I can say here is that Dharmakṣema translates this as "*bu jian zhe wei bu jian foxing* 不見者謂不見佛性", which differs from that in our surviving Sanskrit fragment, in his classical Chinese rendering, no matter what underlying term or phrase he read.[70] The Sanskrit fragment states "the deed of the Tathāgata will not end (or destroyed) unless all beings involved in transmigration recognize the ultimate enlightenment. At that moment, the Buddha will come to complete final *nirvāṇa*. Along with the final *nirvāṇa* (*atyantaparinirvāṇa*), the Buddha will become changeable and absent, like fire and a lamp." Alternatively, the classical Chinese translation by Dharmakṣema states that "*bu jian shui zhi suozuo* 不見誰之所作 (not seeing whose deeds were carried out)" means ignoring *tathāgata*'s deeds. Although Buddha explained *foxing* for beings, *icchantika*s cannot recognize *foxing* due to their transmigration. Thus, it is called "*bu jian rulai suozuo* 不見如來所作 (not seeing that carried out by the Tathagata)." Seeing that the Tathāgata has gone into the ultimate *nirvāṇa*, the *icchantika*s mistakenly thinks that the Tathāgata is impermanent, just like the light that goes out when the oil is exhausted. Specifically, in the Sanskrit fragment, "*dhātu*" does not appear in this passage. On the contrary, Dharmakṣema and his collaborators translated something as "*foxing*," which is the Chinese translation term of "*buddhadhātu*" or "*dhātu*" in many cases.

Evidently, Dharmakṣema and his collaborators translated something, compared with *kalyāṇakṛta* and *saṃmyaksaṃbodhi* found in the extant Sanskrit fragment, as *foxing* in the *Da banniepan jing*. The case that this kind of translation was made by Dharmakṣema's hand is doubtful. Furthermore, the material corresponding to the sentence "*yichanti jian yu rulai bijing niepan*一闡提見於如來畢竟涅槃" cannot be found in the Sanskrit fragment. Accordingly, compared to Dharmakṣema's translation, it is difficult to identify the direct evidence that *icchantika* can also achieve buddhahood in the existing Sanskrit fragments.

Notably, as the above section, the difference between the Sanskrit fragment and Chinese translation is located in the ninth fascicle of the *Da banniepan jing*, which is attributed to a rendering by Dharmakṣema himself. According to previous research, after the finishing of the translation of the first twelve fascicles of the *Da banniepan jing*, Dharmakṣema stayed in Guzang姑臧 and learned the Chinese language for three years.[71] In other words, the section discussed above, where the difference in *foxing* between Sanskrit and Chinese appears, was translated by Dharmakṣema when he was not proficient in the Chinese language. For this reason, it would be understandable if his collaborators and disciples inserted some personal views, or removed agency from Dharmakṣema, into their translations.[72]

According to the *Gaoseng zhuan*高僧傳 [Biographies of Eminent Monks], Dharmakṣema was engaged in the translation work of the *Da banniepan jing* from 414 to 421.[73]

Dharmakṣema intended to go abroad because there was a shortage in the original text of the *Mahāparinirvāṇa-mahāsūtra*. However, due to his mother's death, he had no choice but to stay for several years. After that, he actually went to Khotan and found the middle portion of the *Mahāparinirvāṇa-mahāsūtra*. Dharmakṣema then returned to Guzang and translated it. Finally, he sent people to Khotan and found the latter portion of the *Mahāparinirvāṇa-mahāsūtra*. This was translated into the thirty-three fascicles of the *Da boeniapan jing*. This translation work was launched during the third year of Xuanshi玄始 and finished on the twenty-third of October of the tenth year of Xuanshi, namely the second year of Yongchu永初.

> 讖以《涅槃經》本品数未足，還外国究尋。值其母亡，遂留歳餘。後於于闐更得經本中分，復還姑臧譯之。後又遣使于闐尋得後分，於是續譯為三十三卷。以偽玄始三年初就翻譯，至玄始十年十月二十三日三裟方竟，即宋武永初二年也。[74]

Furthermore, according to the Guanding灌頂 (561–632)'s record, from 414 to 416, Dharmakṣema was engaged in translating the *Da banniepan jing* by collaborating with Zhimeng智猛 (?–452).[75]

When Dharmakṣema arrived at the western Liang state西涼州, Juqu Mengxun沮渠蒙遜 dominated the Longhou隴後 area and the Xuanshi玄始 reign began. During the third year of Xuanshi, Juqu Mengxun asked Dharmakṣema to translate the *Da banniepan jing*. Dharmakṣema translated five fascicles of its original text into twenty fascicles of the classical Chinese translation. After that, due to the shortage in the original text, Juqu mengxun sent people abroad and found eight fascicles. They were the chapters of *bingxing*病行品, *shengxing*聖行品, *fanxing*梵行品, *yingerxing*嬰兒行品, *dewang*德王品, *shizihou*師子吼品, *jiashe*迦葉品 and *chenru*陳如品. Dharmakṣema translated them into twenty fascicles and spread them across northern China. During the fifth year of Xuanshi, the translation work of the *Da banniepan jing* was complete.

> 到西涼州，值沮渠蒙遜割拠隴後，自号玄始。其号三年，請曇無羅讖共猛訳五品，得二十卷。遂恨文義不圓，再遣使外国，更得八品。謂病行、聖行、梵行、嬰兒行、德王、師子吼、迦葉、陳如等品。又翻二十卷，合成四十軸，伝於北方。玄始五年乃得究訖。[76]

If these documents are examined together, they record that Dharmakṣema clearly translated the original text of the *Mahāparinirvāṇa-mahāsūtra* with Zhimeng and the support of other collaborators at least twice.[77] If so, the significant difference in *foxing* between the extant Sanskrit fragments and Dharmakṣema's *Da banniepan jing* discussed in this section belongs to the first twelve fascicles of his translation, which was translated by Dharmakṣema and his collaborators when he had not yet mastered the Chinese language.

Accordingly, the role of Dharmakṣema's collaborators and disciples in his translation of the *Da banniepan jing* is important. Needless to say, it is also likely that Dharmakṣema had seen Faxian's translation, and its use of *foxing*, and himself reasoned that this was a good way of communicating what the *Mahāparinirvāṇa-mahāsūtra* was teaching.

Regarding my hypothesis mentioned above, one of my reviewers once noted: "Even if Dharmakṣema was not familiar with the Chinese language at that time, in my opinion, a translator's limited proficiency in the target language does not make it more likely that the translator would insert into his translation something that is not there in the source language. Probably the reverse argument can also be made, namely, a translator who has excellent proficiency in the target language would then be more likely to insert into his translation something that was not there in the source language. Alternatively, I don't think it makes much sense for the author to suggest that, since his Chinese language was not good enough, then his collaborators and disciples inserted some personal views into their translations. Were Dharmakṣema's colleagues to add anything into the translation of the *Da banniepan jing*, this inserted notion must had already become quite popular before the text was translated." I have to add something of my response towards this query here. In my view, it is likely that either Dharmakṣema or his collaborators and disciples translated some other terms as the Chinese term *foxing*. There are further two possibilities. One is that they created the term *foxing* here. Another possibility is that they used the term *foxing*, which had already become popular before the translation of the *Da banniepan jing*. For the second possibility, when I say Dharmakṣema's collaborators and disciples inserted some personal views into their translations, needless to say, it is also possible that their personal views had been influenced by some terms which had already become popular then. In other words, I do not think that there is a fundamental contradiction between my reviewer's hypothesis and that of mine, although both of our views are merely assumptions.[78]

Additionally, some sections of the *Da banniepan jing* translated by Dharmakṣema state that *icchantika*s also possess *foxing*.

For those who are *icchantika*s, although they possess buddha-nature, they are held down by the stain of their innumerable transgressions, unable to get free, like silkworms inside of cocoons. Because of their karmic conditions, they cannot produce a marvellous cause that would lead to *bodhi* and instead transmigrate through *saṃsāra* with no end in sight.[79]

> 彼一闡提雖有佛性，而為無量罪垢所纏，不能得出。如蠶所繭。以是業緣，不能生於菩提妙因。流轉生死，無有窮已。[80]

As mentioned above, Chinese translators used the term *foxing* to correspond with various original terms. In this section, they translate "*bi yichanti sui you foxing*彼一闡提雖有佛性." Furthermore, although the Chinese translation states that *icchantika*s can merely float in the stream of birth and death without becoming free from transmigration, according to its interpretation, *icchantika*s definitely possess *foxing*. This statement has strongly influenced even wider East Asian Buddhist thought.

Conversely, the *Nihuan jing* translated by Faxian states the following.

*Icchantika*s are separated from the *rulai xing*如來性 (lineage of the *tathāgata*) forever due to committing the crime of criticizing the Buddhist Dharma. It is like the cocoon created by bugs which controls the bugs themselves, so do *icchantika*s. They cannot stimulate their origins of Bodhi in the lineage of the *tathāgata*, so that they cannot become free from transmigration during all lives.

> 彼一闡提於如來性所以永絕，斯由誹謗作大惡業。如彼蠱虫綿網，自纏而無出處。一闡提輩亦復如是。於如來性不能開發起菩提因，乃至一切極生死際。[81]

Evidently, the *rulai xing*如來性 found in this section of the *Nihuan jing* corresponds to the *foxing* stated in the *Da banniepan jing*.[82] However, the *Nihuan jing* clearly states that *icchantika*s are forever separated from the *rulai xing* due to committing the crime of criticizing the Buddhist Dharma. Although Faxian also used the term *foxing* in the *Nihuan*

*jing*, there were fewer uses of the term than those in the *Da banniepan jing*. In this section, Faxian used the term *rulai xing* instead of *foxing*.

Regardless, it remains necessary to confirm whether the original Sanskrit text of the *Nihuan jing* and that of the *Da banniepan jing* are identical or not. From the perspective discussed above, on the relationship between *icchantika* and *rulai xing* (or *foxing*), it seems likely that the assertions of Dharmakṣema's Chinese collaborators also influenced the translation more or less.[83] Identical to the classical Chinese translation by Dharmakṣema, the Tibetan translation of the *Mahāparinirvāṇa-mahāsūtra* states that Buddha-nature is also within the *icchantika*s' bodies; among various translations of the *Mahāparinirvāṇa-mahāsūtra*, the only version that states the *icchantika* without Buddha-nature is the *Nihuan jing*. It is very likely that the translators of the Tibetan translation consulted Dharmakṣema's classical Chinese translation.[84]

According to Ōchō, the *Nihuan jing* denies the possibility that *icchantika*s can achieve buddhahood, whereas the *Da banniepan jing* states that *icchantika*s can achieve this if they successfully see their *shanxin*善心 (good mind).[85] The discussion above also clarifies Ōchō's assertion. The sentence "*icchaṃtikāḥ kalyāṇakṛtaṃ na*" in the extant Sanskrit fragment corresponds to "*bu jian zhe wei bu jian foxing*不見者謂不見佛性." "*Icchantika*s fail to see virtuous deed (*kalyāṇakṛta*)" in this Sanskrit fragment corresponds to "*icchantika*s can achieve buddhahood" in the *Da banniepan jing*. While we cannot know what Dharmakṣema was seeing in his Sanskrit text, I contend that this translation strongly supported the theory that *icchantika*s can achieve buddhahood in East Asian Buddhism.[86]

Concerning this issue, Takasaki notes that the *Da banniepan jing* translated by Dharmakṣema, alongside the *Nihuan jing* translated by Faxian, also states that *icchantika*s do not possess *foxing* before the eleventh fascicle. In contrast, after this fascicle, the *Da banniepan jing* admits the possibility that *icchantika*s could finally achieve buddhahood. Saliently, the above portion can only be found in the classical Chinese translation rather than the Sanskrit or other texts.[87] Furthermore, Matsumoto Shirō松本史朗 asserts that in the *Da banniepan jing*, the *you foxing*有佛性 (possessing Buddha-nature) does not mean *jie chengfo*皆成佛 (accomplishing buddhahood for all beings).[88] Both Takasaki and Matsumoto were aware of the difference between the first twelve and subsequent fascicles of the *Da banniepan jing*. Clearly, their assertions reinforce my opinion.

To summarize, as a classical Chinese Buddhist canon text translated at the beginning of the fifth century, the *Da banniepan jing* used the term *foxing*, which cannot be correspondingly confirmed in the surviving Sanskrit fragments of the *Mahāparinirvāṇa-mahāsūtra*. *Foxing* was most naturally translated *buddhadhātu*, but the Sanskrit fragments do not mention *buddhadhātu*. Those sections where the difference between the Sanskrit and the Chinese translation of *foxing* appears belong to the translation made by Dharmakṣema before he was proficient in the Chinese language.[89] For this reason, it is not impossible that his collaborators and disciples may have inserted some personal views into their translations. It is possible that his inserted notion had already become popular before the translation of the *Da banniepan jing*. Meanwhile, it is also likely that Dharmakṣema had seen Faxian's translation, and its use of *foxing*, and himself reasoned that this was a good, shorthand way of communicating what the *Mahāparinirvāṇa-mahāsūtra* was teaching.

Moreover, over half of the *Da banniepan jing*, after the first 10 *juan*, is unique to this version. We have no Sanskrit fragments corresponding to its content and no Tibetan, apart from a Tibetan translation made from Dharmakṣema's Chinese translation.[90]

## 5. *Foxing* in the *Da fangdeng wuxiang jing* 大方等無想經 and Guṇabhadra's Renderings

In this short section, as a supplement for this study, I intend to slightly mention the *Da fangdeng wuxiang jing* 大方等無想經 [Skt. *Mahāmeghasūtra*; T387] by Dharmakṣema and the renderings by Guṇabhadra 求那跋陀羅 (394–468) from roughly the same time.

As is mentioned in the first section of this article, according to the *Chu sanzang ji ji*出三藏記集, 11 texts were regarded as Dharmakṣema's translations. It is impossible to

analyze the term *foxing* in all of them in this space-limited article. Since we have a Tibetan translation of the *Da fangdeng wuxiang jing*, the *Sprin chen po'i mdo* (Derge no. 232; Peking no. 898), I merely have a look at one case of *foxing* in this text.

The *Mahāmeghasūtra* is a *tathāgatagarbha* doctrinal sutra, overlapping with the *Mahāparinirvāṇa-mahāsūtra*. The *Da fangdeng wuxiang jing*, the Chinese rendering of the *Mahāmeghasūtra*, was also translated by Dharmakṣema. In the *Da fangdeng wuxiang jing* and the corresponding Tibetan translation, we can find the following example:

| *Da fangdeng wuxiang jing*大方等無想經 (Taishō no. 387, 1102b2–3) | *Sprin chen po'i mdo* (Derge no. 232, 194b) |
| --- | --- |
| 猛風起者，喻如來常。風入毛孔者，喻諸眾生悉有佛性。 | de bzhin du 'dir yang ting nge 'dzin gyis de bzhin gshegs pa'i yon tan rtag pa nyid kyi yon tan gyis bsgos pa'i rlung nyon mongs pa'i nam mkha' la ldang bar byed cing/ |

In the Tibetan rendering, confirming a reasonable corresponding term to the Chinese term *foxing* here is a little difficult. We cannot find *de bzhin gshegs pa'i snying po* (buddha-nature), but only *de bzhin gshegs pa'i yon tan* or *rtag pa nyid kyi yon tan*. We also confirm some similar cases in the *Da fangdeng wuxiang jing* like this.[91] In other words, Dharmakṣema uses the term *foxing* in the *Da fangdeng wuxiang jing*, while the corresponding term or phrase in the Tibetan translation is unclear.

Although the *Da fangdeng wuxiang jing* may appear to be a partial translation of the *Mahāmegha-sūtra*, the real situation was more likely that the original text was still incomplete when Dharmakṣema brought it to China.[92]

Guṇabhadra was born in central India to a brāhmaṇa family and departed from Sri Lanka for China, arriving in Guangzhou by sea in around 435.[93] He translated some famous Mahāyānist sutras, including the *Yangjuemoluo jing*央掘魔羅經 [Pāli. *Aṅgulimāla-sutta*; Skt. *Aṅgulimālīyasūtra*; T120] and the *Da fagu jing* 大法鼓經 [Skt. *Mahābherīhārakasūtra*; T270], in which the term *foxing* can be found. The *Da banniepan jing* translated by Dharmakṣema, however, was brought to Jiankang建康, present-day Nanjing南京, becoming the foundation of the southern version of this scripture's Chinese rendering. Huiyan 慧嚴 (363–433), Huiguan 慧觀 (4th to 5th centuries CE) and Xie Lingyun 謝靈運 (385–433) edited this scripture into the southern version in 436.[94] Moreover, with their help, Guṇabhadra translated some texts.[95] For this reason, the term *foxing* found in Guṇabhadra's renderings, which were translated later than the renderings translated by Faxian, Buddhabhadra and Dharmakṣema, was probably more or less influenced by the *Da banniepan jing*.[96]

## 6. The Interpretations of *Foxing* in Later Chinese Buddhism

Dharmakṣema was proficient at incantation and respected in many countries.[97] Finally, he was assassinated by Juqu Mengxun沮渠蒙遜 (368–433), the King of Northern Liang北涼. Northern Wei destroyed Northern Liang very soon afterwards. Dharmakṣema's disciples and collaborators moved to Pingcheng平城, the capital of Northern Wei北魏. Furthermore, the *Da banniepan jing* translated by Dharmakṣema was brought to Jiankang建康, becoming the foundation of the southern version of this scripture's classical Chinese translation. Hence, in my opinion, the translations and concepts in Dharmakṣema's system strongly influenced Buddhism during the Northern Wei period, especially Bodhiruci菩提流支 (6th century CE; active in China after 508) and Ratnamati勒那摩提 (6th century CE; active in China after 508). The *Da banniepan jing*, which was sufficiently researched in Northern Wei, became the foundation of the doctrines of the Dilun tradition地論宗, including Huiyuan of the Jingying temple淨影寺慧遠 (523–592).[98] The influence of the *Da banniepan jing* on Chinese Buddhist thought is apparent. As abundant amount of research already exists on this issue[99]; I will, therefore, merely discuss the cases of Huiyuan of the Jingying temple and Guanding of the Tiantai tradition天台宗 in this section.

In the *Da banniepan jing yi ji*大般涅槃經義記 [Meaning of the Great Nirvana Sutra], Huiyuan's commentary on the *Da banniepan jing*, he states the following.

There is a type of Buddha-nature that the *icchantika*s have but those who possess wholesome roots do not. [Namely, the *icchantika*s] have the unwholesome nature, and hence they lack wholesome nature. Due to dependent origination [based upon] the Buddha-nature, unwholesome aggregates arise. Hence unwholesome aggregates are named Buddha-nature, which the *icchantika*s have. There is another type of Buddha-nature that those who possess wholesome roots have but the *icchantika*s don't. Those who have advanced above the first [bodhisattva-]stage are called people with wholesome roots. Or more broadly, the bodhisattvas above the stage of buddha-gotra (Chin. *zhongxing di* 種性地) are named wholesome human beings (i.e., people with wholesome roots). They have wholesome nature and lack unwholesome nature. There is a type of Buddha-nature that both [of the above two groups of people] have, namely, they both have the nature as the principle (*lixing* 理性is a short form for *li foxing* 理佛性). There is another type of Buddha-nature that both [of the above two groups of people] do not have, namely, neither of them has the nature as the result (meaning that they have not attained Buddhahood).

> 或有佛性，一闡提有，善根無者，有不善性，無其善性。佛性緣起為不善陰，故
> 不善陰名為佛性。闡提有此。或有佛性，善根人有，闡提無者，初地已上名善根
> 人，通則種性已上菩薩斯名善人。彼有善性，無不善性。或有佛性，二人俱有，
> 俱有理性。或性，二俱無，俱無果性。[100]

Huiyuan's interpretation contains vital information. The statement that *icchantika*s also possess Buddha-nature (*foxing*) is clearly influenced by the *Da banniepan jing*. According to Huiyuan's explanation, *icchantika*s have the arising Buddha-nature and principal Buddha-nature. Among these two, the arising Buddha-nature is only possessed by *icchantika*s. Thus, it is clear that Huiyuan was deeply influenced by the *Da banniepan jing* and regarded it as the foundation of his theory of Buddha-nature.[101] The most significant connection in this passage is the term *foxing*, which cannot be correspondingly confirmed in the current Sanskrit fragments of the *Mahāparinirvāṇa-mahāsūtra*.

Accordingly, after Northern Wei extinguished Northern Liang, the *Da banniepan jing* translated by Dharmakṣema, and his assistants' interpretations, were likely conveyed to Pingcheng, the Northern Wei capital, and influenced the Dilun monastic tradition, including monks such as Huiyuan.

Moreover, Guanding states in the *Da banniepan jing shu* 大般涅槃經疏 [Commentary on the *Mahāparinirvāṇa-Sūtra*] as below.

First, hearing (Chin. *wen* 聞) is divine ear (Skt. *divyaśrotra*; Chin. *tianer* 天耳). Seeing (Chin. *jian* 見) is divine eye (Skt. *divyacakṣus*; Chin. *tianyan* 天眼). They relate to the *jishen tong* 即身通 (penetrating understanding with the body). Second, the ninth stage is hearing, in which one can see Buddha-nature. The tenth stage is sight, in which one can complete and clarify himself through seeing Buddha-nature. Achieving the ninth stage by liberation of wisdom is the particular hearing which is manifested without normal hearing. Achieving the tenth stage from the ninth stage is the particular seeing which is manifested without normal seeing. Achieving the buddhahood stage from the tenth stage is the particular achieving which is manifested without normal achieving.

> 一云，聞即天耳，見即天眼，至即身通。二云，九地為聞，見佛性，十地為眼。
> 見佛性，具足明了。今因慧解脫至第九地，是不聞而聞。因九地至十地，即不見
> 而見。因十地至佛地，為不至而至。[102]

Guanding mentions the term "*jian foxing* 見佛性" as found in the *Da banniepan jing* translated by Dharmakṣema in his commentary and states that one would see *foxing* if he has achieved the ninth stage of bodhisattvas' stages. As has been discussed above, Dharmakṣema and his assistants translated something, which is reported as the *gotra* of *tathāgata* (*rigs*) and good deeds (*kalyāṇakṛta*) in the surviving Sanskrit fragments of the *Mahāparinirvāṇa-mahāsūtra*, as *jian foxing* (seeing Buddha-nature).[103] Evidently, the *gotra* of *tathāgata* (*rigs*) means those people who will or have achieved the boundary of the *tathāgata*. However, Guanding only used the term *jian foxing*, while *zhongxing* (Skt. *gotra*;

lineage/caste) cannot be found. Furthermore, he attempted to integrate *jian foxing* with the theory of the stages of the bodhisattvas, especially the ninth stage.[104]

Evidently, the *Da banniepan jing* was translated by Dharmakṣema with his assistants' interpretations, influenced not only the Dilun monastic tradition, but also the Tiantai monastic tradition and monks such as Guanding. The term "*jian foxing*" found in the *Da banniepan jing* was emphasized by Guanding, the direct disciple of Zhiyi智顗 (538–597).[105] As is well known, this term greatly influenced the later Tiantai tradition, the Huayan華嚴 (Jp. Kegon) tradition and Chan禪 (Jp. Zen) Buddhism through some of the early Tiantai monks such as Guanding. According to Whalen Lai, the Tiantai tradition, based on the *Lotus Sūtra* (Chin. *Fahua jing*法華經), superseded the Nirvāṇa tradition涅槃宗 by incorporating many of its ideas.[106] We can therefore imagine the wide influence of the *Da banniepan jing* (*Nirvāṇa Sūtra*) and the Nirvāṇa tradition. Needless to say, the most important idea of the Nirvāṇa tradition is the theory of *foxing*.[107] However, also as discussed above, the extant Sanskrit fragments of the *Mahāparinirvāṇa-mahāsūtra* states that *icchantika*s do not see good deeds (*kalyāṇakṛta*). Instead, the term or phrase in this corresponding place was rendered as "*bu jian foxi*ng不見佛性" in the *Da banniepan jing*. The most important term *foxing* cannot be found as a fixed term in our current Sanskrit fragment.

While probably correct from the perspective of those Indic original texts, I suppose that a crucial point exists. That is, from the perspective of a Chinese reader, in all these cases there is only one single term—the Chinese word *foxing*佛性.[108]

## 7. Conclusions

In East Asian Buddhism, *rulaizang* (Skt. *tathāgatagarbha*) is sometimes considered a synonym of *foxing* (Buddha-nature) because the relationship between these two terms was ambiguous in Chinese Buddhism since some monks and schools declared that *foxing* is the same as *rulaizang*. The early translators who emphasized some translated terms as *foxing* were Buddhabhadra and Dharmakṣema, two Indian Buddhist monks living in China in the first half of the fifth century. That is to say, the cases of the Chinese term *foxing* appeared during the Northern Liang dynasty (397–439) and the second half of the Eastern Jin (317–420) are probably the key to probing some early cases where the term *foxing* appeared.

The *Da fangdeng rulaizang jing* (Skt. *Tathāgatagarbha-sūtra*) translated by Buddhabhadra is a very early classical Chinese Buddhist canon text where the term *foxing* is clearly used to express Buddha-nature. However, the Chinese term *foxing* is difficult to confirm in Amoghavajra's classical Chinese translation. Although a lack of clarity remains about Buddhabhadra's motivation, as an early classical Chinese Buddhist canon text, the *Da fangdeng rulaizang jing* used the term *foxing*, which cannot be confirmed in other extant translations of the *Tathāgatagarbha-sūtra*.[109]

Compared to the *Da fangdeng rulaizang jing*, the *Da banniepan jing* (Skt. *Mahāparinirvāṇa-mahāsūtra*) translated by Dharmakṣema has exerted a much greater influence on Chinese Buddhist thought. As another early classical Chinese Buddhist canonical text, the *Da banniepan jing* also used the term *foxing*, which cannot be correspondingly confirmed in the surviving Sanskrit fragments of the *Mahāparinirvāṇa-mahāsūtra*.[110] The sections where the differences between the Sanskrit fragment and the Chinese term *foxing* appear belong to Dharmakṣema's early translation before he was proficient in the Chinese language.

Furthermore, it is very unrealistic to believe that the same person or group simultaneously translated all forty fascicles of the *Da banniepan jing*. Different people would have edited these fascicles in several stages. Notably, *buddhadhātu*, the original Sanskrit term of the Chinese term *foxing*, which is regarded as the most significant term in the *Da banniepan jing*, cannot be found in the extant Sanskrit fragments of this scripture. Dharmakṣema translated the original text of the *Mahāparinirvāṇa-mahāsūtra* with Zhimeng and the support of other collaborators at least twice. The significant difference between the Sanskrit fragments and the classical Chinese translation of the *Mahāparinirvāṇa-mahāsūtra* in this article belongs to the first twelve fascicles of Dharmakṣema's translation aided by his disciples

and collaborators when he had not yet mastered the Chinese language. Therefore, we should not ignore the role of his assistants. Of course, it is also likely that Dharmakṣema had seen Faxian's translation and its use of *foxing* and, himself, reasoned that this was a good, shorthand way of communicating what the *Mahāparinirvāṇa-mahāsūtra* was teaching.

Meanwhile, we frequently find *sangs rgyas kyi khams/dbyings*, which is a translation of *buddhadhātu* in the Tibetan rendering of the *Mahāparinirvāṇa-mahāsūtra*. This leads us to presume that Faxian and Dharmakṣema both read versions of the *Mahāparinirvāṇa-mahāsūtra* that used this term and translated it and other terms, including those I mentioned in this article, with *foxing*.

It is likely that Faxian translated a version of the *Mahāparinirvāṇa-mahāsūtra* that featured *buddhadhātu* as *foxing*. Buddhabhadra, in the same period, translated a version of the *Tathāgatagarbha-sūtra*. In some passages, he had favoured the term *foxing* over a literal translation of the Sanskrit. As a contemporary monk with Buddhabhadra and Faxian, Dharmakṣema translated the *Mahāparinirvāṇa-mahāsūtra*, going further than Faxian by using the term *foxing* regularly. Our Sanskrit fragments of the *Mahāparinirvāṇa-mahāsūtra* are surely of a later date.[111] We can suspect that both Dharmakṣema and Buddhabhadra employ *foxing* as a non-literal translation, after Faxian.

Moreover, after Northern Wei extinguished Northern Liang, the *Da banniepan jing* translated by Dharmakṣema, and the interpretations of his collaborators and disciples were likely conveyed to Pingcheng, the Northern Wei capital. These two texts translated by Buddhabhadra and Dharmakṣema respectively, especially the *Da banniepan jing*, deeply influenced the Dilun monastic tradition. Among these, the term *foxing* and its Sinicism explanations played a highly significant role, influencing the whole of East Asian Buddhist thought. Needless to say, the controversies focusing on the concept of "Buddha-nature" within all sentient beings in East Asian Buddhism, including the theory of *tathāgatagarbha*, are closely related to the term *foxing* and its Sinicism explanations discussed in this article. However, it is difficult to clarify the accurate origin of the Chinese term *foxing* at least at the beginning of the fifth century in the relevant Sanskrit and Tibetan fragments and texts at present.[112]

The aim of this article was not to be exhaustive or comprehensive but to provide some additional reflections on the term *foxing* represented in the *Da fangdeng rulaizang jing* and the *Da banniepan jing*, two contemporary classical Chinese renderings, suggesting possible further research. Although it is a little difficult to say that the *Da fangdeng rulaizang jing* and the *Da banniepan jing* are the earliest classical Chinese Buddhist canon texts where the term *foxing* is clearly used to express Buddha-nature, these two Chinese renderings are very early-stage translations in this sense. It is hoped that this study can make a small contribution to reconsider the origin and background of the Chinese term *foxing* within the historical context of Chinese Buddhist translation.

**Funding:** This research was funded by Robert H. N. Ho Family Foundation Postdoctoral Fellowship in Buddhist Studies 2019.

**Data Availability Statement:** Not applicable.

**Conflicts of Interest:** The author declares no conflict of interest.

## Notes

1    Regarding these controversies on Buddha-nature and *tathāgatagarbha*, see Swanson (1993).

2    The term *buddhadhātu* was also translated with *foxing* in some texts. While Sanskrit fragments of the *Mahāparinirvāṇa-mahāsūtra* do not preserve *buddhadhātu*, the Tibetan corresponding to the Chinese preserves *sangs rgyas kyi khams/dbyings*, which is a rendering of *buddhadhātu*. See Jones (2020b). Versions of the *Ratnagotravibhāgavyākhyā* confirm that *foxing* was used to translate *buddhadhātu*. However, following Radich (2015), Dharmakṣema seems unlikely to have made a direct translation '*buddhadhātu > foxing*' in his work. See Radich (2015, pp. 23–24).

3    Concerning the development of Nirvāṇa tradition in China, see Fuse (1974a, 1974b); Mather (1981).

[4] We also find the term *foxing* in the *Mohe bore boluomi jing*摩訶般若波羅蜜經 translated by Kumārajīva鳩摩羅什 (344–413) (*T*. 223: 8.299a23-24) and the *Dazhidu lun*大智度論 (T. 1509: 25.499a21-22). Since the Sanskrit text of Kumārajīva's *Larger Prajñāpāramitā* is extant and edited, further work on the comparison with Sanskrit text is inevitable. According to most of the previous research, however, it is very likely that Kumārajīva did not know the theory of Buddha-nature.

[5] Regarding this approach, see Radich (2015); Zimmermann (2002); Jones (2021).

[6] The Sanskrit *Mahāparinirvāṇa-mahāsūtra* is a way to identify this as the Mahayana *Mahaparinirvana-sutra*.

[7] The term *dhātu* itself means other things also. The range of things communicated by *dhātu* is not perfectly covered by the character *xing* 性. Concerning the meaning of the word *xing* in Chinese non-Buddhist culture, see Satō (1998).

[8] Concerning the history of the Buddha-nature concept in Chinese Buddhism, there are already a large number of books and articles. For instance, Tokiwa (1930); Liu (1982); Lai (1988); and Liu (2008), etc. However, most of these researches hardly considered and used the relevant Sanskrit and Tibetan texts.

[9] *Chu sanzang ji ji*出三蔵記集 2, T. 2145: 55.11b10–25.

[10] Regarding the original name and its translation of Tanwuchen (Tanmochen) 曇無(摩)讖, see Fuse (1974a, 1974b, pp. 116–38).

[11] Concerning the subsequence of the texts translated by Dharmakṣema, Chen Jinhua陳金華 has further research. See Chen (2004). The Indian Buddhist Missionary Dharmakṣema (385–433): A New Dating of his Arrival in Guzang and of his Translations. *T'oung Pao* 90(4): 215–63. Chen argues that Dharmakṣema in fact performed no translation until 421.

[12] Regarding this fact, see Fuse (1974a, pp. 98–99).

[13] Regarding the biography of Faxian, see Legge (1886); Adachi (1940); Zhang (1985).

[14] Radich demonstrates that the *Mahāparinirvāṇa-mahāsūtra* attributed to Faxian (*T*7) and the translation of the *Buddhacarita* (*T*192) attributed to Baoyun寶雲 (376?–449) are closely related and were probably both translated by Baoyun. See Radich (2019b).

[15] According to Lettere, the *Chu sanzang ji ji* played a role in limiting the impact of Baoyun's translation activities. Moreover, Huijiao慧皎 (497–554) attempted to blame Baoyun's poor interpreting, rather than Buddhabhadra's contrast with Kumārajīva, as the cause of the contrasts between Buddhabhadra and the *saṅgha* in Chang'an長安. See Lettere (2020).

[16] Regarding the biography of Dharmakṣema, see Chen (2004) and Stephen Hodge (2012).

[17] Regarding this report, see Takasaki (1987).

[18] Concerning this work, see Matsuda (1988).

[19] See the following works, Habata (2009), Habata (2013).

[20] These fragments had been edited, see Habata (2019).

[21] Ōno (1954, pp. 236–37). Also see Michael Radich's database of attributions (https://dazangthings.nz/cbc/text/1323/, accessed on 1 June 2022).

[22] Regarding this, see Ōchō (1981, p. 39).

[23] Concerning Feng's statement, see Feng (1976).

[24] Chen's argument had been accepted by many scholars, see Chen (2004, pp. 215–63).

[25] Most recently, Radich considers the material exclusive to Dharmakṣema's translation. See Radich (2019a).

[26] The following book can be related to the relationship between the *Da fangdeng rulaizang jing* and the *Da banniepan jing*: Matsumoto (2021), chapter 3, Nyoraizō kyō to nehan gyō 『如来蔵経』と 『涅槃経』.

[27] *Fozu tongji*佛祖統紀 36, T. 2035: 49.342b15–343a3.

[28] For instance, the *Mahāparinirvāṇa-mahāsūtra* was translated by Dharmakṣema (T374) and edited by Huiguan慧觀 (T375). Kumārajīva met Buddhabhadra and Vimalākṣa (Chin. Beimo luocha卑摩羅叉) in Changan長安. After Kumārajīva's death, Vimalākṣa left Changan for Jiangling江陵 and cooperated with Huiguan. See the *Lidai sanbao ji*代三寶記 7, T. 2034: 49.70c22–71a1.

[29] Although a decision to treat the *Fozu tongji* as an historical source for the early fifth century on a part with the primary documents needs to be further discussed, the information recorded here mentions some accurate dates and persons, which can be consulted as at least some subsidiary materials. Meanwhile, I concede that the *Fozu tongji* is a much later source, which has its disadvantages and limitations.

[30] Regarding this interpretation, see Michael Zimmermann (2002, pp. 39–50); Jones (2020b, p. 145); Jones (2020a); Kanō (2020).

[31] The *Lidai sanbao ji* 代三寶紀 records: "大方等如來藏經一卷 (元熙二年於道場寺出, 是第二譯, 見道祖晉世雜錄, 與法立出者小異。) . . . 右一十五部一百一十五卷, 安帝世, 北天竺國三藏禪師佛馱跋陀羅, 晉言覺賢。" (T49, no. 2034, 71a13-b1) This indicates that the *Da fangdeng rulaizang jing*大方等如來藏經 (*T* vol. 16, no. 666), translated by Buddhabhadra, is one of the two Chinese renderings of the *Tathāgatagarbha-sutra*.

[32] Strickmann writes: "Properly speaking, many of [Amoghavajra's 167 'translations'] were not translations at all. Instead, they might better be called 'adaptations'; essentially, he refurbished them in line with his own terminology and ritual practice. This becomes even more striking in those cases where texts 'translated' by Amoghavajra are known to have been written in China centuries earlier, and directly in Chinese. A substantial part of Amoghavajra's output thus comprises revisions of books already known in China, rather than new materials. Among the remaining, a good many cannot be found either in corresponding

Sanskrit manuscripts or in Tibetan translation—at least not in the form in which Amoghavajra presents them." See Strickmann (2002). Also see Michael Radich's database of attributions (https://dazangthings.nz/cbc/text/967/, accessed on 1 June 2022).

33   Zimmermann (2002) suggests two recensions of the text of the *Tathāgatagarbhasūtra*: TGS1 represented just by Buddhabhadra's version and TGS2 represented by other three extant versions. See Zimmermann (2002, pp. 12–17).

34   Concerning the bibliography of the *Tathāgatagarbha-sūtra*, see (1958). *Kanzō sanyaku taishō nyoraizō kyō* 漢蔵三訳対照如来蔵経. Kyoto: Bukkyōbunka kenkyūjo 仏教文化研究所.

35   Regarding this argument, see Zimmermann (2002, p. 7).

36   *Da fangdeng rulaizang jing* 大方等如來藏經 1, T. 666: 16.457b28-c8.

37   *eṣā kulaputra dharmāṇāṃ dharmatā/utpādād vā tathāgatānām anutpādād vā sadaivaite sattvās tathāgatagarbhā iti* / (Johnston 1950, 73, pp. 11–12).

38   This is a citation from Zimmermann's translation, see Zimmermann (2002, pp. 103–6).

39   *Da fangdeng rulaizang jing* 大方等如來藏經 1, T. 666: 16.459a10-13.

40   This is a citation from Zimmermann's translation, see Zimmermann (2002, pp. 136–38).

41   *Da fangdeng rulaizang jing* 大方等如來藏經 1, T. 666: 16.458b6-10.

42   This is a citation from Zimmermann's translation, see Zimmermann (2002, p. 119).

43   Following Ichikawa, it can be assumed that some possible underlying terms are related to the classical Chinese term *rulaizang* 如來藏 through the extant Tibetan translation. They are: *tathāgatagarbha*; *tathāgatadharmatā*; *dharmatā*; *buddhatva*; *sattva*; *sugatakāya*; *jinakāya*; *buddhakāya*; *tathāgatagotra*; *jinaputra*; *tathāgatatva*. See Ichikawa (1982).

44   Concerning this, see King (1995).

45   As the newest research attempting to explain this problem in the context of Indian religions, see Jones (2020a).

46   This is based on Habata's work, see Habata (2015).

47   Regarding this, see Takasaki (1974).

48   Concerning this statement, see Shimoda (1997); Michael Radich (2015).

49   Regarding this argument, see Kanō (2017).

50   According to one of my anonymous reviewers of this article, however, here the Tibetan version is invaluable: *sangs rgyas kyi khams/dbyings* very probably rendered *buddhadhātu*, and this corresponds to *foxing* in Dharmakṣema's and Faxian's versions. I am grateful to my reviewer for this reminder.

51   According to Radich, both Chinese translations of the *Mahāparinirvāṇa-mahāsūtra* frequently feature terms such as *foxing* 佛性 and *rulaixing* 如來性. These terms may not obviously look like translations or equivalents for *tathagatagarbha*. See Michael Radich (2015, p. 23).

52   According to Habata Hiromi, the Sanskrit original of *Mahāparinirvāṇa-mahāsūtra* has come down to us only in fragments, while the underlying Sanskrit term of the Chinese term *foxing* 佛性 and its intended meaning poses difficulties. Moreover, it is very likely that Dharmakṣema preferred the word *foxing* in his translations, independent from the existing Sanskrit text. See Habata (2015, pp. 176–96).

53   *Second Bhāvanākrama*, Peking ed., No. 5311, A 49a8-49b3, sDe dge ed., No. 3916 Ki 45a5-6.

54   *Da banniepan jing* 大般涅槃經 30, T. 374: 12.547a9-11.

55   Regarding this, see Yoshimura (1974, pp. 381–82).

56   Concerning Matsuda's argument, see Matsuda (1988, pp. 13–14).

57   Following one of my anonymous reviewers of this article, this material in the *Da banniepan jing* comes from content exclusive to that version, for which we have no known Indic basis. It is likely that Kamalaśīla here exhibits knowledge of Dharmakṣema's translation of the *Mahāparinirvāṇa-mahāsūtra*. The alternatives to this scenario are: (a) Kamalaśīla knew the Tibetan translation of Dharmakṣema's Chinese into Tibetan (Derge no. 119)—but this was only in the eleventh century. (b) Kamalaśīla knew an Indic version of the material translated by Dharmakṣema. I think the alternative (b) is more likely, namely, that both Kamalaśīla and Dharmakṣema were following a hitherto unknown Sanskrit version of the *Mahāparinirvāṇa-mahāsūtra*.

58   The Sanskrit passages in this article is based upon Johnston (1950), see *Ratnagotravibhāga*, ed. Edward Hamilton Johnston. Patna: The Bihar Research Society, 1950.

59   *Jiujing yisheng baoxing lun* 究竟一乘宝性論 3, T. 1611: 31.828b1-5.

60   Regarding this issue of the *Ratnagotravibhāga*, see Li (2016).

61   The Chinese version of the *Ratnagotravibhāga* is often pretty different to the Sanskrit and Tibetan. We cannot rule out that *tathāgatagotra* was not seen by Ratnamati.

62   Indeed, the fact remains that the *Jiujing yisheng baoxing lun* elsewhere clearly also used *foxing* to render Skt. *buddhadhātu*, not only Skt. *gotra*. In other words, both Dharmakṣema and Ratnamati came to use the term *foxing* to translate a broader range of terms and phrases, including, needless to say, *tathāgatagarbha* and *buddhadhātu*.

63  Mizutani analyzes the origin of icchantika in his work, see Mizutani (1965).

64  Concerning the relationship between icchantika and buddha-nature in East Asian Buddhism, see Tokiwa (1930).

65  Habata (2019) renders this sentence as the following German translation: "*Die Icchantikas, 'eine heilvolle Tat nicht sehend', 'sieht' (sehen) aber die 'bo-se' (d. h.) tadelhafte 'angeklagte' (Tat). 'sukṛta (gute Tat) bedeutet 'Erwachen'.*" See Habata (2019, p. 154).

66  This is my translation from the Sanskrit fragment. Habata (2019) renders this section as the following German translation: "*Zu jener Zeit, wenn die Lebewesen, nachdem sie einmütig geworden sind, zum höchsten vollkommenen Erwachen erwachen werden, zu dieser Zeit wird der icchan- tika, auch wenn er böse ist, erwachen. Er sieht zu dieser Zeit das höchste Erwachen, (nämlich) die gute Tat nicht. Erkenne so, du Erfahrener! Wessen Tat sieht er nicht? (Die Tat) des Tathāgata. Zu jener Zeit (in der Zukunft), wenn alle Lebewesen, die in den Saṃsāra gekom-men sind, zum höchsten vollkommenen Erwachen erwachen werden, zu dieser Zeit wird die Tat des Tathāgata nicht erschöpft sein. (Trotzdem behauptet der Icchantika) so etwas wie 'Nachdem der Buddha durch das vollständige Parinirvāṇa vollkom- men zur Ruhe gelangt ist, wird er nicht mehr anwesend sein, wie eine Lampe, wie ein Feuer aufgrund des Aufgebrauchtseins des Brennholzes.' (Dies ist) das böse, tadelhafte und angeklagte Karma des Icchantika.*" See Habata (2019, pp. 157–58).

67  Matsuda (1988, pp. 45–46). In addition, Habata Hiromi edited the extant Sanskrit fragments and provided a new transla- tion in 2019, which is more in-depth than that of Matsuda. In Habata (2019), this passage is as the following: "icchaṃtikāḥ kalyāṇakṛtaṃ na paśyataḥ paśyati tu pāpaṃ ni- <n>di{śi}taṃ garhitaṃ ca yaḥ (r6) sukṛ[t](a)ṃ (b)o[dh]i[r] ity arthaḥ na vyaiti na gacchatīty arthaḥ sandheti kalyāṇam ity arthaḥ saṃndhākarma viśiṣṭakalyāṇaṃ ka- sya nāgacchati bhadrakarma icchaṃtikasya nāgac[ch]a(r7)ti (kasya nāgacchati kuśa)- la{ṃ}satva icchaṃtikā iti [v]i(śrutā) + + + + …+ + …[k]iṃ mūlāgaṃ sūtrapratikṣe- paḥ tasmād bhetavyaṃ sūtrapratikṣepako hi dāruṇaṃ (v1) ta(smād bibhyati paṇḍit)[ā] : dhīrā mahāpathai saṃti [s]aṃ[sk](ā)[r](ā) + + + + + + … (na bibh)y(a)ti gacchaṃti goraṃ mānavaśaṃ tato nāsādayaṃti durmedha- saḥ taṃ ca<ndra>m uddha(v2)raṃta {;} iv(a) v[ā]narā āsādayaṃti [t]u [p]aṇḍitā dhīrā nar[e]ndrā iva mahāpathe kaḥ kṛtaṃ na paśyati <icchaṃtikaḥ> saṃsārakoṭyāṃ sa na paśyati arthaṃ bhāṣiṣye: saṃkṣepasamu(v3)ccayaṃ tasmād bhetavya[m pa]rama- dāruṇā[t]* yadā sar[vbasa]tvā eka- manaso bhūtvā anuttarāṃ saṃmyaksaṃbodhim abhisaṃ- botsya<ṃ>te ; tadā icchaṃtika [p]ā(v4)po <'>pi saṃ[bo]tsyate sa tadā parāṃ bodhi su{rvba}kṛ(ta)[ṃ] na paśyati ; evaṃ jāṃṇīṣva viśārada kasya kṛtaṃ na paśyati ; tathāgatasya yadā sarvbas(a)(v5)tvā anuttarāṃ saṃmyaksaṃbodhim abhisaṃbotsya<ṃ>te saṃsāra[g]a[t]ā tadā tathāgatasya kṛtaṃ na vinakṣya(t)[i t]adā parinirvbāyātyaṃtapa-rinirvbāṇena ; a[n]i(v6)tyo buddho [bh]a[v]iṣyati ; [d]īpa ive[ndh]anak[ṣ]ayād agnir iva ta[dv]at* icchaṃtikasya pāpa[ṃ] karma garhitaṃ nindita(ṃ) ca." See Habata (2019, pp. 154–58). Comparing Matsuda (1988) with Habata (2019), it seems that there is no significant difference here about the underlying expressions of the Chinese term *foxing* in Dharmakṣema's translation.

68  This is based on Blum's translation, see Blum (2013).

69  *Da banniepan jing* 大般涅槃經 9, T. 374: 12.418b28-c26.

70  It is difficult to find corresponding terms or phrases here in both Tibetan and Faxian's translations. See Radich (2015, p. 189).

71  Concerning this, see Chen (2004, pp. 215–63).

72  According to the preface in the eighth fascicle of the *Chu sanzang ji ji*, the Sanskrit text related to the first ten fascicles of the *Da banniepan jing* translated by Dharmakṣema had been brought to China by Zhimeng 智猛 (?–452). The "*Da niepan jing ji di shiqi*" 大涅槃經記第十七 in the eighth fascicle of the *Chu sanzang ji ji* states that: "此《大涅槃經》初十卷有五品。其胡本是東方道人智猛從天竺將來，暫憩高昌。有天竺沙門曇無讖，廣學博見，道俗兼綜。遊方觀化，先在燉煌。河西王宿植洪業，素心冥契。契應王公，朝統士衆。西定燉煌，會遇其人，神解悟識。請迎詣州，安止内苑。遣使高昌，取此胡本，命讖譯出。" (T. 2145: 55.60a) That is, although Dharmakṣema is considered the translator of the *Da banniepan jing*, this classical Chinese translation version and its Sanskrit original text are closely related to the Western Regions of China.

73  As mentioned above, Chen argues that Dharmakṣema in fact made no translation until 421. See Chen (2004). Although I quote some materials from the *Gaoseng zhuan* here, I accept Chen's conclusion.

74  *Gaoseng zhuan* 高僧傳 2, T. 2059: 50.336b1-6.

75  Regarding the references to Dharmakṣema and Zhimeng in the *Gaoseng zhuan*, see Naoumi (1986).

76  *Da banniepan jing xuanyi* 大般涅槃經玄義 2, T. 1765: 38.14a26-b2.

77  On the contrary, as mentioned in the first section, Chen Jinhua argues that Dharmakṣema in fact made no translation until 421. See Chen (2004).

78  On the other hand, it is also important to realise the textual fluidity of Sanskrit original of the *Mahāparinirvāṇamahāsūtra*, in addition to the possibility of the translator's creation or insertion. Accordingly, it looks that there is currently no clear witnesses to ascertain whether the translation term *foxing* is the translator's faithful translation of the Sanskrit original, the translator's creation, or his insertion.

79  This is based on Blum's translation, see Mark L. Blum (2013, p. 287).

80  *Da banniepan jing* 大般涅槃經 9, T. 374: 12.419b5-7.

81  *Foshuo da bannihuan jing* 佛説大般泥洹經 6, T. 376: 12.893a8-11.

82  Concerning this paragraph, the Tibetan translation states: " 'dod chen pa rnams la yang de bzhin gshegs pa'i snying po yod mod kyi (Even the *icchantika* has *tathāgatagarbha*.)/ 'on kyang g-yogs ma shin tu stug par 'dug go//dper na dar gyi srin bu rang nyid kyis kun nas dkris te/sgo ma btod pas phyir 'byung mi nus pa de bzhin du/de bzhin gshegs pa'i snying po yang de'i las kyi

nyes pas 'dod chen pa'i khong nas dbyung bar mi nus so//de bas na 'khor ba'i mtha' las byang chub kyi rgyu mi 'thob bo //" See Habata (2013, p. 349).

[83]  Although it is likely that the original Sanskrit text of the *Nihuan jing* and that of the *Da banniepan jing* were not identical, as mentioned above, Dharmakṣema came to use the term *foxing* to translate a broader range of terms and phrases. We also should not totally deny the element of the activity of translators.

[84]  Dharmakṣema's version is closer to the Tibetan translation. In other words, Faxian's rendering looks to be the exception on this matter.

[85]  Concerning this argument, see Ōchō (1981, p. 42).

[86]  The extant Sanskrit fragments of the *Mahāparinirvāṇa-mahāsūtra* currently available to us is just a part of the entire text. I must confess that there might be other examples that contradict to my argument in undiscovered portions of the Sanskrit original text.

[87]  Takasaki points out this in his work, see Takasaki (1983).

[88]  Matsumoto points out this in his work, see Matsumoto (1989).

[89]  The remaining content of Dharmakṣema's translation is unique to that version, so it is difficult to assess how close it is to other versions of the *Mahāparinirvāṇa-mahāsūtra*. See Radich (2019a) for the most recent discussion of this material. It looks like the Dharmakṣema-unique material is something of a compilation of material from various sources, from Central Asia or plausibly the work of himself. I am grateful to one of my anonymous reviewers for reminding me of this.

[90]  Concerning this, see Radich (2019a) and Jones (2020a).

[91]  To my knowledge, Christopher Jones is researching this issue in the *Da fangdeng wuxiang jing*. I look forward to his forthcoming publication. Regarding his previous research, see Christopher Jones (2016).

[92]  Regarding this, see Ono and Maruyama (1937, pp. 486–87).

[93]  See *Gaoseng zhuan* 高僧傳 2, T. 2059: 50.344a18-26.

[94]  See *Kaiyuan shijiao lu* 開元釋教録 11, T. 2154: 55.591a2-5.

[95]  See *Gaoseng zhuan* 高僧傳 3, T. 2059: 50.344a5-b10.

[96]  Although the example in this section does not reflect *sangs rgyas kyi khams/dbyings*, Tibetan versions of the works by Guṇabhadra reflect *sangs rgyas kyi khams/dbyings*. There is cause to believe that he was translating *buddhadhātu* in some other places.

[97]  See *Gaoseng zhuan* 高僧傳 2, T. 2059: 50.335c16-337b4.

[98]  Ten fascicles of the *Niepan yi ji* 涅槃義記, written by Huiyuan 慧遠 of the Jingying temple, currently exist. This is the only extant complete commentary on the *Da banniepan jing* translated by Dharmakṣema.

[99]  Concerning this issue, see Fuse (1974a, 1974b); Richard B. Mather (1981, pp. 155–73).

[100]  *Da banniepan jing yi ji* 大般涅槃經義記 9, T. 1764: 37.873b27-c4.

[101]  Regarding Huiyuan's interpretation of *foxing*, see Keng (2013).

[102]  *Da banniepan jing shu* 大般涅槃經疏 22, T. 1767: 38.169b29-c4.

[103]  Concerning *gotra* in the context of this literature, see David Seyfort Ruegg (1976).

[104]  There is no agreement in the extant Sanskrit materials as to the exact nature of these bodhisattva stages. See Har Dayal (1932).

[105]  Regarding Zhiyi's attitude toward Buddha-nature, see Paul Swanson (1990).

[106]  Lai notes this in his work, see Lai (1982).

[107]  For example, Sengrui 僧叡 (378–444) stated that the *Lotus Sūtra*'s concept of the Buddha's omniscience anticipated the *Da banniepan jing* (*Nirvāna Sūtra*)'s idea of *foxing* (Buddha-nature).

[108]  Cf. Robert Sharf's perspective remarks on the role played by translations in Chinese Buddhism. See Sharf (2001, pp. 18–20).

[109]  It will be helpful if there is a comparative table of the term *foxing* and its equivalents Skt. or Tib. of the *Tathāgatagarbhasūtra* in this article. Concerning this, we can consult Zimmermann (2002, pp. 50–52).

[110]  Since the extant Sanskrit fragments are just a small part of the entire text of the Sūtra and there were most probably various versions of Sanskrit originals of this Sūtra, it is difficult to approach a final conclusion currently.

[111]  As hypothesized by Hodge (2012), these Sanskrit and Tibetan materials of the *Mahāparinirvāṇa-mahāsūtra* show signs of redaction. It is possible that instances of *buddhadhātu* were replaced with *tathāgatagarbha*.

[112]  It is a fact that we find supporting evidence in other Tibetan works where *sangs rgyas kyi khams* (Skt. *buddhadhātu*) corresponds to Chin. *foxing*. Meanwhile, according to one of my anonymous reviewers, some Tibetan renderings were sometimes translated from Chinese, instead of Sanskrit texts. It might be still a complex issue even if we find a completed Sanskrit text due to their chronological relationship.

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
