# Peer review of "A Study of the Early-Stage Translations of Foxing佛性 in Chinese Buddhism: The Da Banniepan Jing大般涅槃經 Trans. Dharmakṣema and the Da Fangdeng Rulaizang Jing大方等如來藏經 Trans. Buddhabhadra"

_religions, doi:10.3390/rel13070619_

Round 1

Reviewer 1 Report

The research topic is significant. The author did a good discussion on it. But there are some problems with the writing style, language used, and format:

1. Compared with other sections, the introduction is too brief. The author should construct the literature review in this section, which includes previous scholars' work to highlight this article's research questions and hypothesis.

2. The language used and writing style should be improved. It seems like the article is translated from another language (such as Chinese) to English. Some of the sentences are hard to understand. In addition, I suggest the author shorten and summarize the article to make the presentation clear and smoothen the sentences so the outsiders could understand easily.

3. The citation format is wrong. It should be changed to APA style.  For additional notes, it should be put after the main text.

4. In the conclusion, the author should elaborate more on the research findings.

5. It is not suitable to quote too many texts from such Buddhist classics in the article, and no need to show it in different languages. The author should focus on his/her opinions and elaborate with the supportive text.

Author Response

Dear my reviewer,

Thank you for your review and very useful suggestions. I am sorry for dropping a long submission to you within so short review duration. I will try my best to revise this submission according to your comments.

>1. Compared with other sections, the introduction is too brief. The author should construct the literature review in this section, which includes previous scholars' work to highlight this article's research questions and hypothesis.

  I will add some previous research to highlight this article in the introduction section.

>2. I suggest the author shorten and summarize the article to make the presentation clear and smoothen the sentences so the outsiders could understand easily.

  As you pointed out, there are some long narrative sentences and repeated passages in this submission. I will try to shorten some of them.

>3. The citation format is wrong. It should be changed to APA style.  For additional notes, it should be put after the main text.

  I will change the format as you mentioned.

>4. In the conclusion, the author should elaborate more on the research findings.

  I will focus on the most important findings in the conclusion.

>5. It is not suitable to quote too many texts from such Buddhist classics in the article, and no need to show it in different languages.

  Let me try to shorten some passages of Buddhist classics.

Appreciate your review and suggestions again. It must waste your time. I will revise the submission as soon as possible to provide a new revised version.

Best regards,

Author of the submission

Reviewer 2 Report

Line 34:

“containers for tathāgataˮ:

This understanding is based on the compound analysis of genitive tatpuruṣa, and this translation is fine as a genitive tatpuruṣa. But more natural interpretation of tathāgatagarbha in the expression “sarvasattvās tathāgatagarbhāḥˮ is bahuvrīhi (“those who contain tathāgataˮ), although the Ratnagotravibhāga offers the three kinds of compound analysis of tathāgatagarbha. 

 page 7, fn. 26:

The meaning of the Chinese is to be clarified more specifically. It will be helpful for readers if the author introduces previous studies on the meaning of the word in Chinese non-Buddhist texts (and Buddhist texts) in this and preceding period.

Line 392:

Kamalaśīla (ca. 740–797) quotes the following in the Bhāvanākrama

Better to specify which Bhāvanākrama it is (I or II or III?).

Line 396, 405, etc.:

samadhi > samādhi

Lines 602–641:

A reviewer's argumentation mentioned here (in lines 602–641) seems to be more plausible. The author's defend written here seems to be weak. 

 On the other hand, it is also important to point out the textual fluidity of Sanskrit original of the Mahāparinirvāṇamahāsūtra (in addition to the possibility of the translator's creation or insertion).

Accordingly, it looks that there is currently no clear witnesses to ascertain whether the translation-term foxing is the translator's faithful translation of the Sanskrit original, the translator's creation, or his insertion.

Lines 702–704:

It is better to add that the extant Sanskrit fragments of the Mahāparinirvāṇamahāsūtra currently available to us is just a tiny part of the entire text. There might be many examples that contradict to the author's argument in undiscovered portions of the Sanskrit original. In this regard, the Sanskrit fragments hardly work as a strong witness to confirm the author's argument.

"Conclusion" part:

It will be helpful for readers if the author present a comparative table of the term foxing and its equivalents Skt. or Tib. of the Tathāgatagarbhasūtra in a simple manner (not an extensive one), if possible.

  I agree the importance of the term foxing in the Chinese translation of the Mahāparinirvāṇamahāsūtra, but the lack of its corresponding Sanskrit buddhadhātu in the extant Sanskrit fragments is not to be excessively emphasized, because the extant Sanskrit fragments are just a small part of the entire text of the Sūtra and because there were most probably various versions of Sanskrit originals of this Sūtra.

The following book can be related to the issue dealt with in this paper: Shirō, Matsumoto, Bukkyōshisōhihan, Kyoto: Hōzōkan, 2021.

Author Response

Dear my reviewer,

Thank you for your review and very useful suggestions. I am sorry for dropping a long submission to you within so short review duration. I will try my best to revise this submission according to your comments.

I will try to amend all places you noted. Furthermore, the book you suggested at the end will also be consulted and mentioned in the revised version.

Appreciate your review and suggestions again. It must waste your time. I will revise the submission as soon as possible to provide a new revised version.

Best regards,

Author of the submission

Reviewer 3 Report

This essay on Buddha nature (faxing) is informative, with a clear and logical argument relying on numerous languages, on a very important topic for East Asian Buddhism. I do not have many comments with regard to the academic content, which is sound, but offer some editorial suggestions (which may or may not be accepted by the author and/or the publisher.

Line 17 and many others: the author uses the title “Mahaparinirvana-mahasutra” and I assume there is a reason for this (although there are some places where it uses only Mahaparinirvana-sutra). I suppose this is a way to identify this as the Mahayana Mahaparinirvana-sutra, and there are places (see below) where the author translates 大乗 as just “maha” instead of “Mahayana”. Perhaps a note could clarify this issue?

Line 24: for the Keywords, faxing, Da banniepan jing, and Da fangdeng rulaizang jing should be in italics.

The notes give detailed bibliographical information for each cited text, but this information is available in the list of References and could (should?) be abbreviated in the notes. For example, the reference in note one could be just “see Swanson 1993; note 2 just “See Jones 2020” and “See Radich 2014” and so forth. The details explaining “T” in note 6 are not necessary. Everyone who reads this article will be familiar with the convention “T” for the Taisho canon, and also the details are given in the References.

Note 4: One major problem: the contents of note 4 do not seem to have anything to do with the placement of this note; the note should be about Buddhabhadra and Dharmaksema but refers to two translations by Kumarajiva. Perhaps the contents of the note were copied or substituted by mistake, or a note deleted?

Line 70: “Tranlsation” should be “Translation”

The Chinese characters are given for modern authors, but since this information is available in the References they do not need to be included in the text. I suggest deleting the kanji for names such as Fuse Kōgaku (line 100), Takasaki Jikidō (line 127), Matsuda Kazunobu (line 128), Habata Hiromi (line 129), Ono Hōdō (line 132), Ōchō Enichi (line 137), Chen Jinhua (line 143), and so forth…

Note 9, there are two references by Fuse Kogaku for 1974, so if the information in the notes is abbreviated to Kogaku 1974, these should be identified as 1974a or 1974b.

Lines 121-122: “Ono Hodo thinks more reliable the preface (…) by Hexi Daolang ****.” would be better as “Ono Hodo thinks the preface (…) by Hexi Daolang **** is more reliable.”

Lines 155-156: Instead of “At last” it would be better to say “Finally,”

Line 169: 迦維衞 should be “Kapilavastu”, not “Kapilabastu”?

Line 176: “Huiyuan ** (334-416)’s disciples” would be better “Huiyuan’s ** (334-416) disciples”.

There are two references to Radich’s work in 2019; these should be identified as 2019a or 2019b.

Line 192: “lack of the comparison” would be better “the lack of comparison”

Note 25: “documents need to be” should be “documents needs to be”, since the verb refers back to “a decision to treat”?

Note 25: “disadvantages and the limitations” should be “disadvantages and limitations”?

Note 26: “is not perfectly that covered” should be “is not perfectly covered”?

Note 27: “Tokiwa Dijō” should be “Tokiwa Daijō”; this is correct in the References.

Lines 196-198: I suggest rewording this to “Conversely, many scholars in Chinese Buddhist Studies have hardly used the relevant Sanskrit and Tibetan…” Note 27 names a number of scholars, but all of them a bit old. I think more recently scholars (such as Matsumoto) have used the relevant Sanskrit and Tibetan in discussion Buddha nature.

Note 28: there are two references by Jones in 2020; these should be identified as 2020a and 2020b.

Lines 355-356: “In this section, the problem is that which term…” would be better “This section focusses on which term…”

Lines 366-367: “Moreover, there is no the case that…” should be “Moveover, there is no example where…”

Line 688: “was looking at his” should be “was seeing in his”.

Line 749: “Moveover, with the help of them,…” would be better “Moreover, with their help,…”

Line 751-752: change to “was probably more or less influenced by…”

Lines 762 and 763: change “acted in” to “active in”

Line 805: as a translation for 即身通, I suggest “penetrating understanding with the body” instead of “connection with the body”. (By the way, here [lines 803-824] I think Guanding is following the analysis by Zhiyi in, for example, the 摩訶止観).

Line 915: change “Reference” to “References”

Line 943: add Chinese characters to name of Chen Jinhua?

Lines 955-963: add Japanese characters to name of Habata Hiromi?

There are four references by Habata 2013, 2019, 2015, 2009), but they are not in dated order. These, and places where other authors have more than one reference, should be standardized by date, either ascending (2009 first and 2019 last) or descending (2019 first and 2009 last).

Lines 971-976: the four references by Jones should be listed in either ascending or descending order, and the two for 2020 identified as 2020a and 2020b.

English middle initials should be in correct order: “Jones, Christopher V.” not “Jones, V. Christopher”; “Sharf, Robert H.” not “Sharf, H. Robert”; “Swanson, Paul L.” not “Swanson, L. Paul”.

Line 1001: The title of Matsuda’s book has 大乗涅槃経; shouldn’t this be “the Mahayana Mahaparinirvana-sutra” and not “Mahaparinirvana-mahasutra”?

The information for Ruegg 1976 in note 99 is not listed in the References.

Line 1048: again, the title of Yoshimura’s book has大乗仏教思想; shouldn’t this be “Mahayana Buddhist Thought” and not “Maha Buddhist 

Author Response

Dear my reviewer,

Thank you for your review and very useful suggestions. I am sorry for dropping a long submission to you within so short review duration. I will try my best to revise this submission according to your comments.

I will try to amend all places you noted, and add the references you reminded in the revised version.

Appreciate your review and suggestions again. It must waste your time. I will revise the submission as soon as possible to provide a new revised version.

Best regards,

Author of the submission